# The flashfm approach for fine-mapping multiple quantitative traits

N. Hernández [1], J. Soenksen [2,3], P. Newcombe[1], M. Sandhu[4], I. Barroso [2], C. Wallace [1,5] & J. L. Asimit [1✉]

Joint fine-mapping that leverages information between quantitative traits could improve accuracy and resolution over single-trait fine-mapping. Using summary statistics, flashfm (flexible and shared information fine-mapping) fine-maps signals for multiple traits, allowing for missing trait measurements and use of related individuals. In a Bayesian framework, prior model probabilities are formulated to favour model combinations that share causal variants to capitalise on information between traits. Simulation studies demonstrate that both approaches produce broadly equivalent results when traits have no shared causal variants. When traits share at least one causal variant, flashfm reduces the number of potential causal variants by 30% compared with single-trait fine-mapping. In a Ugandan cohort with 33 cardiometabolic traits, flashfm gave a 20% reduction in the total number of potential causal variants from single-trait fine-mapping. Here we show flashfm is computationally efficient and can easily be deployed across publicly available summary statistics for signals in up to six traits.

[1] MRC Biostatistics Unit, University of Cambridge, Cambridge, UK. [2] Exeter Centre of Excellence for Diabetes Research (EXCEED), University of Exeter Medical School, Exeter, UK. [3] School of Life Sciences, University of Glasgow, Glasgow, UK. [4] Dept of Epidemiology & Biostatistics, School of Public Health, Imperial College London, London, UK. [5] Cambridge Institute of Therapeutic Immunology & Infectious Disease (CITIID), University of Cambridge, Cambridge, UK. ✉email: jennifer.asimit@mrc-bsu.cam.ac.uk

Genome-wide association studies (GWAS) have been extremely successful in identifying genetic variants that are associated with a wide spectrum of diseases and related traits[1]. Among these findings are many examples of pleiotropy, where a gene affects several phenotypes[2]. This could be due to a shared variant affecting a pathway involved in multiple related phenotypes. Identifying the causal mutations which underlie such findings is key to facilitating translation into new therapeutic targets or elucidating new biological insights. However, this is a complex task. Lead SNPs (those with the most significant $p$-value) are often correlated (are in high linkage disequilibrium—LD) with many other variants in the genome. In addition, lead variants are not necessarily causal and may be detected due to LD with the causal SNP(s). Statistical fine-mapping is therefore needed to refine sets of potential causal variants. The fewer the number of variants identified as potentially causal, the easier it will be to perform downstream functional validation experiments. Here, we focus on three fine-mapping challenges: multiple trait analysis, missing trait values, and related individuals in a cohort.

Bayesian approaches are common in fine-mapping, and use a Bayes' factor (BF) to summarise the evidence of association, either for each SNP under the assumption of a single causal variant, or for each combination of SNPs in the more flexible multiple causal variant setting. In the more general multiple causal variant setting, the BF for a model compares the evidence for a model consisting of a particular set of SNPs that could be causal for a trait to the null model of no causal SNPs[3–5]. These BFs can be calculated for different combinations of causal SNPs and, for a specified prior probability, posterior probabilities (PP) for each causal variant model are easily calculated, modelling the pattern of association within a region. Models may then be prioritised by PPs.

Current fine-mapping methods do not allow missing trait data, so that a portion of the data is disregarded to avoid any missing measurements. Also, fine-mapping methods that make use of GWAS summary statistics (e.g., JAM[4], FINEMAP[5], CAVIARBF[6] SuSiE[7]) assume the specified sample size $N$ relates to independent individuals, whilst the effective sample size after adjustment for relatedness via a linear mixed model, is $<N$. Such inflated $N$ will suggest more certainty than actually present in the sample.

Some methods use SNP annotations to improve fine-mapping resolution. A potential caveat of such approaches is that they depend on the completeness of the annotation tool. This is not an issue for some annotations that are intrinsic to the DNA itself, such as DNA topology[8–10]. However, most other annotation tools may bias results towards the biology/function that we already understand, until the full functional effect of every variant is known. PAINTOR[11] and DAP-G[12] allow for multiple causal variants and integrate either association strength with functional genomic annotation (PAINTOR) or enrichment-based annotations that consider GWAS data from other traits (DAP-G). PolyFUN[13] leverages functional annotations to specify prior probabilities for existing fine-mapping methods. The CaVEMaN[14] method estimates the probability that the lead SNP for an expression trait is causal for that association, and could assist in SNP prioritisation. KnockoffZoom[15] localizes causal variants at multiple resolutions by testing if a phenotype is independent of all SNPs in a LD block, conditional on the others; it requires individual-level data from unrelated individuals.

Jointly fine-mapping multiple traits could give an improvement in fine-mapping accuracy and resolution, analogous to the power increase for multi-trait GWAS, but this is computationally challenging due to the many possible combinations of models (allowing multiple causal variants) between traits, which is not an issue for multi-trait GWAS that involve testing only one SNP for association with multiple traits. For this reason, few methods exist for fine-mapping with more than two traits.

When multiple traits have signals in the same region, colocalization is often used to evaluate how likely the traits share a causal variant. In some methods colocalization includes the fine-mapping step of identifying potential shared causal variants. HyPrcoloc[16] and mcoloc[17] make the simplifying assumption of at most one causal variant for each trait. HyPrcoloc ignores trait correlations and is only able to incorporate trait correlations by adjusting the prior configuration probabilities; the authors show that ignoring this adjustment can reduce power to detect a cluster of colocalised traits. Correlation between traits is not considered by mcoloc, as it requires that all traits are measured in distinct datasets of unrelated individuals. An approach that allows multiple causal variants, eCAVIAR[18], requires that the traits are measured from independent studies. This is because eCAVIAR is designed to assess if there are shared causal variants between a quantitative trait from a GWAS and expression quantitative trait loci (eQTL), which are often available in independent studies.

One approach for jointly fine-mapping signals for quantitative traits from the same study is to limit the combinations of models by assuming all causal variants are shared between traits and allowing heterogeneity in effects, as in fastPAINTOR[19]. Although this greatly reduces the number of models to consider, this does not allow traits to have different sets of causal variants. Although multi-trait fine-mapping is motivated by traits having shared causal variants, the possibility of a single shared causal variant and additional trait-specific causal variants cannot be ignored, as well as the fact that causal variants may actually differ for all traits. An approach that does not make this assumption is (Multinomial Fine-mapping, MFM[20]) which can be used to fine-map multiple related diseases with shared controls. This approach is made computationally tractable by showing that the joint Bayes' factor (BF) for $M$ diseases with shared controls is a function of the individual disease BFs, model complexity and sample sizes.

In this work, we introduce a method for general quantitative multi-trait fine-mapping that solves the issues outlined above, allowing for related individuals and missing trait measurements: flashfm (flexible and shared information fine-mapping). Through extensive simulations we demonstrate that flashfm improves fine-mapping accuracy (mean PP increase of 0.15 for correct model) and resolution (median percentage reduction of the number of potential causal variants ranges from 31 to 11%) over single-trait fine-mapping, when there is at least one shared causal variant. We also show that flashfm gives higher prioritisation to causal SNPs than fastPAINTOR, most notably when traits have multiple causal variants. We subsequently apply flashfm to genetic association signals for 33 cardiometabolic traits measured in 6407 participants from Uganda - the largest GWAS of a single population from Africa[21,22]; within this sample 2907/6407 are at least second degree related. In this dataset, we see several cases where flashfm improves on single-trait fine-mapping, distinguishing between two models with similar levels of support under single-trait fine-mapping and generally producing higher resolution solutions. In particular, the groups of likely causal variants constructed under flashfm are subsets of those from independent fine-mapping, resulting in finer resolution.

## Results

**Flashfm—conceptual framework**. Flashfm uses a similar Bayesian framework (same prior probabilities) to MFM[20]. However, flashfm addresses different statistical challenges than MFM, as we now have multiple quantitative traits measured on the same individuals, so we need to account for correlation between the

traits. In addition, the statistical modelling differs between the methods, as MFM uses a multinomial logistic framework, whereas flashfm is in a multivariate regression framework.

Flashfm uses GWAS summary statistics to jointly fine-map genetic associations for multiple quantitative traits that have partial sample overlap, and allows flexibility for missing measurements and for related individuals. The GWAS for each trait could either be from a single cohort or from a meta-analysis of multiple cohorts, where traits overlap between cohorts, though may not be measured in all cohorts. As the traits are measured on the same individuals, it requires the trait covariance matrix, which may be calculated from an in-sample study or approximated from the GWAS and trait summary statistics[23]. Flashfm does not require an assumption of exchangeable effect sizes when modelling shared genetic architecture across traits. It uses GWAS summary statistics from each trait to fit the joint models, allowing for multiple causal variants for each trait, with no restrictions on shared causal variants between traits. Flashfm shares information between traits by up-weighting joint models (combinations of fine-mapping models across traits) that have a shared causal variant. For each trait, flashfm outputs the top SNP models and the model posterior probability (PP), adjusted for information from the other traits. It also provides the marginal posterior probability (MPP) that a SNP contributes to any model. A schematic diagram of flashfm shows the link with single-trait fine-mapping (Fig. 1).

In single-trait fine-mapping, models are prioritised by posterior probabilities (PP) which are calculated from the pre-specified prior probability and the BF that is calculated from the data. For multi-trait fine-mapping, we first find the PP for joint models then marginalise the joint PPs to get trait-specific PPs that are adjusted for the other traits; when there are two traits, the multi-trait adjusted PP for model $i$ of trait 1, $PP_i^1$, is $\sum_j PP_{ij}^M$, where $PP_{ij}^M$ is the joint PP of the joint model consisting of the configuration of model $i$ for trait 1 and model $j$ for trait 2. In general, for any number of traits, the trait-adjusted PP for a particular model is found by summing over the PPs of all joint model configurations that contain model $i$ for trait 1.

To generate posterior support for fine-mapping models, flashfm needs to calculate the ABF (approximate Bayes' factor) for all possible model combinations across SNPs and traits. We find expressions of the log(ABF) for each of the joint and marginal models by using the approximation based on the Bayesian information criterion (BIC)[24]. If the traits are independent, then the joint ABF of $M$ traits, denoted $ABF^M$, is the product of the marginal ABFs. As the traits are correlated the joint ABF is not a simple expression, and we derive the difference, using the log-scale: $D_M = \log(ABF^M) - \sum_{j=1}^M \log(ABF_j)$, which simplifies to a term that depends on GWAS summary statistics, covariance matrix of the traits, and sample sizes; $D_M$ varies for each model configuration. For $M$ traits, $D_M = -N/2(\log|\widehat{C_M}| - \log|C_M|)$, where $|C|$ denotes the determinant of matrix $C$, $C_M$ is a $M \times M$ matrix with element $(i, j)$ equal to Cov(trait $i$, trait $j$)/Var(trait $i$), and $\widehat{C_M}$ is the approximation of $C_M$. $C_M$ is constant and depends on the trait covariance matrix, whereas $\widehat{C_M}$ is based on the covariance matrix of the residuals specific to each model configuration and is approximated from the GWAS summary statistics, sample sizes, and SNP covariance matrix from a reference panel (Supplementary Information, Section 1.1). This makes the approximation of $ABF^M$ computationally feasible: $\log(ABF^M) = \sum_{j=1}^M \log(ABF_j) + D_M$.

When there are missing data, $D_M$ includes additional terms that account for the individuals that do not have measurements for all traits, using a combinatorial argument. The joint BF is first expressed as the BF for the multiple traits (at a particular model for each trait) on the portion of the sample that have no missing data, then an additional BF is added for each combination of traits with data available, careful to include each individual in only one term (Supplementary Information, Section 1.2).

The prior probability for the joint models includes a term κ that gives more weight to joint models that have a shared causal variant between the traits. For models $M_i$ (trait 1) and $M_j$ (trait 2), with marginal prior probabilities $p_i$ and $p_j$, we denote the joint model as configuration $C_{ij}$ and set the prior probability as $Pr(C_{ij}) = p_i p_j \kappa^{M_i \cap M_j \neq \varnothing} \tau_{ij}$, such that the joint prior is simply the product of the marginal priors when there is no overlap of variants between models, and is otherwise upweighted. The term κ is derived in a combinatorial manner and is identical to that used in MFM[20]. It requires setting a target odds (TO) of the odds for traits not having a shared causal variant compared to having a shared causal variant; setting TO = 1 coincides with a 50/50 chance of shared causal variant(s) and is the setting that we use. When κ = 1, there is no weight for joint models with shared causal variants and the flashfm PP for each model for a given trait is the same as that from single-trait fine-mapping, which we also refer to as independent fine-mapping, as it does not make use of data from other traits. The term $\tau_{ij}$ is a correction factor that anchors the prior probabilities so that the prior probability of traits having particular model sizes is consistent for different values of κ; identical to MFM[20].

This means that rather than calculating the joint BF for each model, the joint PP could be directly calculated from the model PPs from single-trait fine-mapping; e.g. the model PPs from the *.config file of FINEMAP[5] could be input to flashfm. In the case of two traits, the trait-adjusted PP for model $\gamma_I$ of trait 1 is calculated using

$$\Pr(\gamma_i \text{ for trait 1}|\text{Data}) \propto PP_i \left\{ 1 + (\kappa - 1) \frac{\sum_{j:I_{ij}=1} \delta_{ij} \tau_{ij} PP_j}{\sum_j \delta_{ij} \tau_{ij} PP_j} \right\},$$

where $\delta_{ij} = exp(D_{ij})$ uses the value of $D_M$ for $M = 2$ traits at models $\gamma_i$ for trait 1 and $\gamma_j$ for trait 2 (Supplementary Information, Section 1.3). This gains efficiency by making use of previously generated single-trait fine-mapping results. If single-trait fine-mapping results are not available, flashfm includes a function to run an expanded version of JAM[4], that requires either the SNP correlation matrix and allele frequencies or the SNP matrix (Methods). So, flashfm requires the same information needed for single-trait fine-mapping, as well as the trait covariance matrix and results from single-trait fine-mapping.

In flashfm, relatedness is accounted for by making use of GWAS summary statistics from a linear mixed model method such as GEMMA[25] or BOLT-LMM[26] and using the summary statistics to approximate the effective sample size $N_e$ for each trait - $N_e$ is approximated at each SNP and the median of the $N_e$ is taken as the effective sample size for the trait; the Neff function in flashfm does this calculation (Supplementary Information, Section 1.4).

The final results of interest are the top models for each trait, adjusted for the other traits, rather than the PP for a joint model. This means the results can be used comparably to those from single trait fine mapping, but with an expectation of greater accuracy because of the leverage of information from the other traits.

Even borrowing information across traits does not remove the complication of LD. Two or more SNPs in high LD may provide equivalent statistical information to explain any trait. Therefore, in addition to single SNP posterior probabilities, flashfm output also expresses SNP models in terms of groups of such nearly

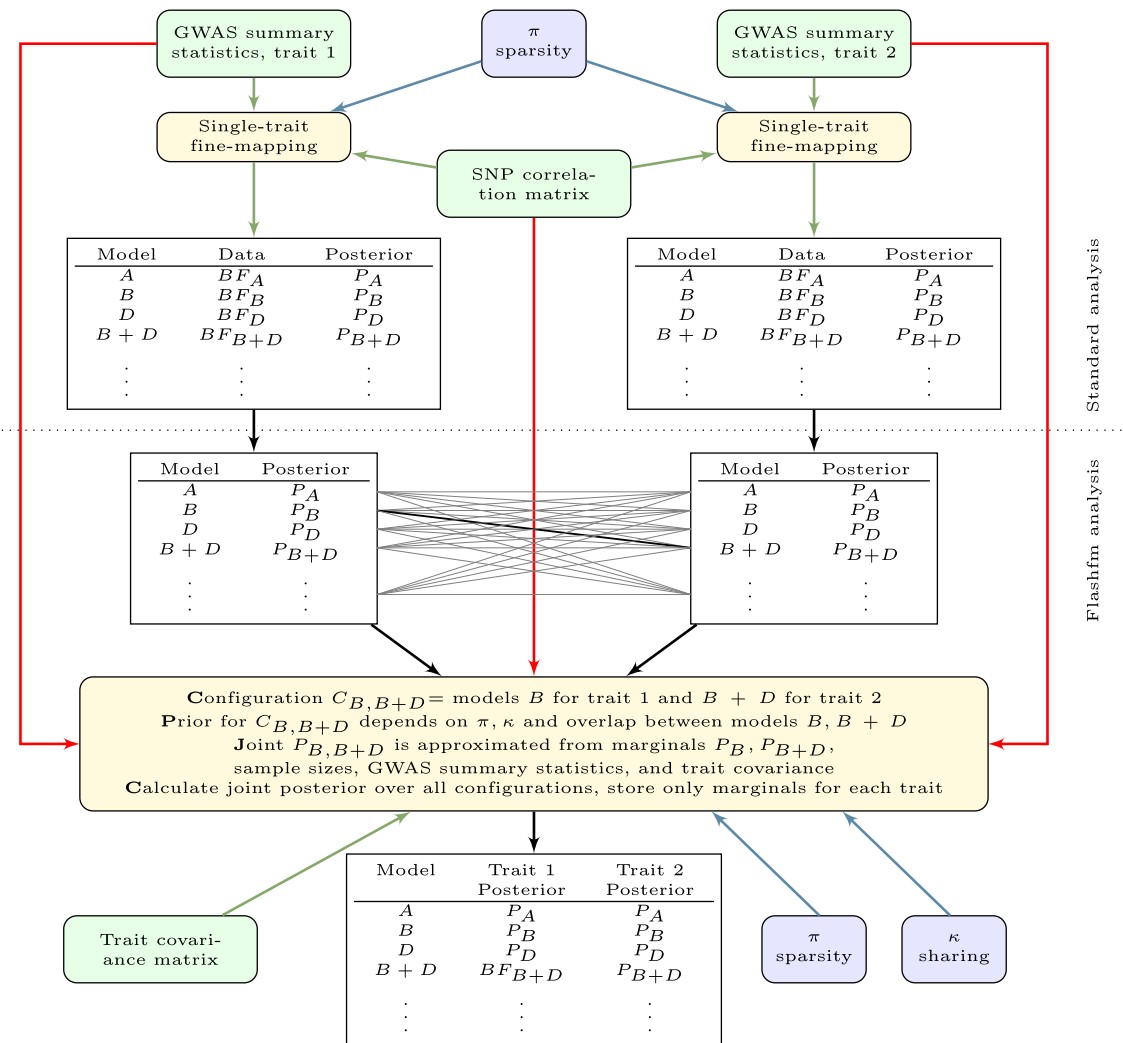

**Fig. 1 Schematic diagram for flashfm.** Flashfm is used for multiple quantitative traits that are measured in the same studies, allowing for missing measurements and family data. First, standard analysis of single-trait fine-mapping is needed for each trait. Then the model posterior probabilities (PPs) from each of these marginal fine-mapping analyses are combined in flashfm, using an approximation to the joint PP, based on an approximation of the joint Bayes' factor. In addition to a SNP correlation matrix, a trait covariance approximation is also needed. Information is shared between traits via a sharing prior that upweights joint models with shared causal variants by a factor of $\kappa$. Memory requirements are reduced by storing only the trait-adjusted marginal PPs for each trait.

equivalent SNPs, constructed so that SNPs in the same group are in LD and rarely appear together in a model (Methods). These SNP groups could be viewed in a similar way to the credible sets that are constructed for single-trait fine-mapping. As flashfm leverages information from the other traits, the per SNP PPs tend to concentrate on fewer SNPs, so the resulting SNP groups tend to be smaller than those from independent fine-mapping.

**Flashfm improves precision over independent fine-mapping.** Extensive simulations of two quantitative traits with varying sample size, proportion of missing trait measurements, and trait correlation suggest the same general conclusion. Flashfm improves precision over independent fine-mapping in terms of higher accuracy, as flashfm gives higher levels of evidence (larger posterior probability) for the correct model (Fig. 2), and finer resolution, as indicated by smaller SNP groups that are constructed based on the PPs from flashfm.

As described in the Methods, according to the specified simulation model, a causal variant was selected from the SNPs $A_1$, $C_1$, and $D_1$, such that one trait had two causal variants $A_1+D_1$,

and the other trait either had two causal variants $A_1+C_1$, one of which was common, or a single distinct causal variant $C_1$ (Supplementary Data 1.1). These SNPs were selected as they are known to represent an example of LD with joint tagging; a different SNP "B" jointly tags $A_1$ and $D_1$, such that when $A_1$ and $D_1$ are causal variants, B is often chosen by fine mapping analyses[20]; this is a difficult region to fine-map due to the potential joint tagging. The data were simulated across 345 SNPs, including many in LD with $A_1$, $C_1$, and $D_1$ (Methods, Supplementary Fig. 1). We label the groups constructed by each fine-mapping approach such that a group that contains $A_1$ is labelled A, and if no group contains $A_1$, but there is a group with a SNP that has $r^2 > 0.7$, then this group is labelled A; likewise for the labelling of groups B, C, D, and J.

To demonstrate that flashfm is well-calibrated, we considered simulations of two traits such that trait 1 has causal variants $A_1$ and $D_1$ and trait 2 has no causal variants in the region. Sample sizes $N = 2000$ and 5000 both give nearly identical results between single-trait fine-mapping and flashfm: the PP of the null model for trait 2 when $N = 2000$ is 0.960 (single-trait) and 0.957 (flashfm) and for $N = 5000$ the null model PPs are 0.966 for both

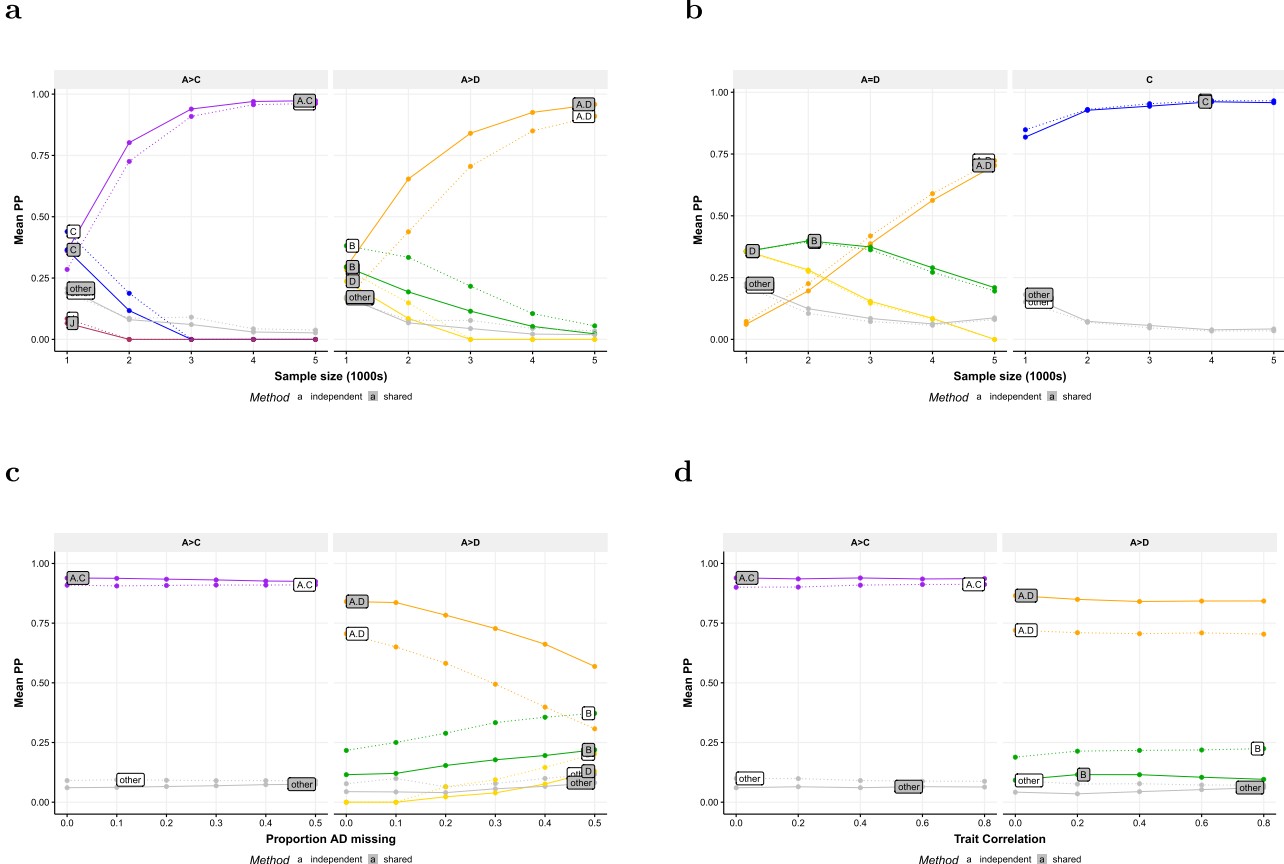

**Fig. 2 Comparison of fine-mapping from flashfm and single-trait analyses.** When traits share a causal variant, flashfm has higher accuracy than single-trait finemapping, regardless of amount of missing data and trait correlation; both methods have similar accuracy when there are no shared causal variants. Causal variants were simulated for two traits with models defined by SNP groups from the IL2RA region. We vary sample size when the traits share a causal variant (**a**) and do not share any causal variants (**b**). At fixed sample size $N = 3000$, we vary the proportion of missing data for one trait (**c**) and vary the trait correlation (**d**). In **a**, **c** and **d** Trait 1 has causal variants A+C, while trait 2 has A+D causal variants, both A causal variants with the same effect size: $\beta_A = log(1.4)$ and $\beta_D = \beta_C = log(1.25)$. In **a** and **b** there are no missing data and the sample size varies from 1000 to 5000. In **c** the sample size is fixed at 3000 and the proportion of missing data for trait A+D varies from 0 to 0.5. In **d** the sample size is fixed as 3000 and the correlation between traits varies. In **b** Trait 1 has causal variants A+D with $\beta_A = log(1.25)$ and $\beta_D = (1.25)$, while trait 2 has a single causal variant C with $\beta_C = log(1.25)$. Results are based on 300 replications. Source data are provided in Supplementary Data 1, Supplementary Data 1.2, 1.3, 1.6, 1.7.

methods. The median difference in SNP group sizes between those constructed from flashfm and those based on independent fine-mapping is zero, and at a given sample size, both methods give the same probabilities that the SNP groups contain the true causal variants: 0.984 (A; $N = 2000$), 0.991 (D; $N = 2000$), 0.996 (A,D; $N = 5000$).

In all settings, for groups C and D, there were negligible differences in group sizes between those constructed from flashfm and those based on independent fine-mapping; groups C and D were not shared between traits so do not gain from shared information. For all settings there is a negligible difference in probability that the SNP group contains the causal variant used in the simulation (i.e., similar coverage) between single and multi-trait fine-mapping. We provide detailed resolution results for group A (Table 1). From this point, the A+D trait refers to a trait simulated to have causal variants $A_1+D_1$.

When there is no shared causal variant (trait 1 has causal variants $A_1+D_1$ and trait 2 has causal variant $C_1$), there are negligible differences between flashfm and independent fine-mapping; the model PPs are nearly indistinguishable (Fig. 2; Supplementary Data 1.2) and for each sample size the median difference in group A sizes between the methods is zero, with similar coverage (Table 1). Thus, application of joint fine mapping where variants are not shared does not incur a penalty.

When both traits have two causal variants, one of which is shared, flashfm tends to show higher support than independent fine-mapping for the true model (mean PP increase of 0.17 with a sample size of at least 3000; Fig. 2, Supplementary Fig. 2, Supplementary Data 1.3–1.4). In low sample sizes independent fine-mapping of the A+D trait either prefers B or has little difference between A+D and B, then as sample size increases to 3000 there is a clear switch to the correct model, A+D. This switch occurs faster for flashfm (at $N = 2000$) due to the borrowed information between the traits. Also, flashfm tends to result in fewer SNPs in group A with a median size reduction of 32% for $N \geq 3000$ (Table 1, Supplementary Data 1.5).

For a sample of $N = 3000$ and varying proportions (0 to 0.5) of missing data from the trait with causal variants A+D, flashfm showed an average gain of 0.20 in the level of PP support for A+D over that of independent fine-mapping (Fig. 2; Supplementary Data 1.6). When half the data were missing for trait A+D independent fine-mapping no longer had a preference for the true model. There is negligible difference between mean PP for the two methods for the trait with A+C causal variants, regardless of the proportion of missing data for the A+D trait. Although flashfm has finer resolution for each degree of missingness, the improvement is highest for lower proportions of missingness (median reduction in group size of 23%) (Table 1).

**Table 1 Resolution comparison between single-trait fine-mapping and flashfm.**

| | Trait 1 (A+D); Trait 2 (A+C); Vary sample size | | | Trait 1 (A+D; Trait 2 (A+C); Vary proportion missing trait 1data | |
| --- | --- | --- | --- | --- | --- |
| N | Median percentage size reduction | Single-trait group A coverage | Multi-trait group A coverage | Trait 1 proportion missing, $p_1$ | Median percentage size reduction |
| 1000 | 0 | 1 | 0.986 | 0 | 28.5 |
| 2000 | 10.5 | 0.997 | 0.99 | 0.1 | 31.4 |
| 3000 | 28.5 | 1 | 1 | 0.2 | 27.8 |
| 4000 | 32.5 | 1 | 0.997 | 0.3 | 18.8 |
| 5000 | 33.3 | 1 | 1 | 0.4 | 16.7 |
| | | | | 0.5 | 10.5 |

| | Trait 1 (A+D); Trait 2 (C); Vary sample size | | | Trait 1 (A+D); Trait 2 (A+C); Vary trait correlation | |
| --- | --- | --- | --- | --- | --- |
| N | Median percentage size reduction | Single-trait group A coverage | Multi-trait group A coverage | Cor($Y_1$, $Y_2$) | Median percentage size reduction |
| 1000 | 0 | 0.944 | 0.944 | 0 | 33.3 |
| 2000 | 0 | 0.98 | 0.97 | 0.2 | 33.3 |
| 3000 | 0 | 1 | 0.99 | 0.4 | 28.5 |
| 4000 | 0 | 0.988 | 0.984 | 0.6 | 21.1 |
| 5000 | 0 | 0.996 | 0.996 | 0.8 | 14.3 |

When traits share a causal variant, flashfm tends to yield smaller SNP groups than those from single-trait fine-mapping, regardless of amount of missing data and trait correlation; both methods have similar resolution and accuracy when there are no shared causal variants. In simulations with a shared causal variant A, (trait 1 is A+D, trait 2 is A+C), $\beta_A = \log(1.4)$ for both traits 1 and 2; trait 1 has a second causal variant D and trait 2 has second causal variant C, both with $\beta = \log(1.25)$. In the non-shared causal variant setting (A+D, C), all causal variants have $\beta = \log(1.25)$. Traits 1 and 2 have correlation 0.4 and were both measured on all individuals, unless otherwise specified. When proportion missing data and trait correlation vary, sample size is 3000. The region has 345 SNPS and was simulated to mimic the LD structure of the IL2RA region, 10p-6030000-6220000 (GRCh37/hg19). Results are based on 300 replications.

**Table 2 Median flashfm running time (with second and third quartiles), in seconds.**

| Number of Traits | 250-SNP Region (67 kb) | 500-SNP Region (144 kb) | 1000-SNP Region (312 kb) |
| --- | --- | --- | --- |
| 2 | 2 (1, 7) | 5 (2, 15) | 5 (1, 16) |
| 3 | 13 (5, 33) | 15 (8, 40) | 16 (5, 59) |
| 4 | 435 (49, 2173) | 583 (116, 1790) | 168 (32, 740) |

Flashfm was run using cpp = 0.99 and single-trait fine-mapping results from JAM, using the extended version (JAMexpandedCor.multi) in the flashfm package. Median time was measured over 100 replications in simulations of 2, 3, and 4 traits having correlation 0.4 and sample size 5000. The regions were subsets of the CTLA4 region 2q-204446258-204816382 (GRCh37/hg19).

By varying the correlation (0 to 0.8) between two traits (A+D and A+C) we found that the PP remained similar within each method and for trait A+D flashfm gave a median PP increase of 0.14 over independent fine-mapping (Fig. 2; Supplementary Data 1.7). Compared to independent fine-mapping, flashfm reduces the median group size 28.5%, with the greatest reduction at lower levels of correlation (33.3% reduction when the trait correlation is 0.2 or 0; Table 1). Low/moderate correlation likely gives higher gains from sharing information between traits because there is more for a trait to "learn" from the other trait; as the traits are measured on the same cohort, highly correlated traits will have similar information to each other so not as much as a gain compared to lower correlations.

**Flashfm is computationally efficient and robust.** We profiled the running time of flashfm, given input from single-trait fine-mapping via expanded JAM that uses the SNP correlation matrix and RAFs (JAMexpandedCor.multi; https://github.com/jennasimit/flashfm/blob/master/R/jamexpanded.corX.R), varying the number of SNPs in a region and varying the number of traits in simulated data with 100 replications in each setting. All simulations were done within a region containing CTLA4 (Methods, Supplementary Fig. 3) and we provide the median running times, as well as second and third

quartiles (Table 2). For all three region sizes, flashfm tends to run in under one minute when there are two or three traits; at four traits, flashfm tends to run in under 10 min.

Within a given region size, as expected, the time increases with the number of traits. However, there was not an observed increase in time as the region size increases. The region that we continuously reduced initially contained 1231 SNPs and was previously defined for fine-mapping of autoimmune diseases[20]. JAMexpandedCor.multi involves first running JAM single-trait fine-mapping considering multi-SNP models with tag SNPs ($r^2 = 0.99$), and then expanding these models to include tagged SNPs by interchanging tag SNPs with their tagged SNPs within each model. As the prior probabilities depend on the number of SNPs in a region, the region of 1000 SNPs tended to run faster (4 traits median 168 s) than the smaller regions (4 traits medians 435 s and 583 s for 250 and 500 SNPs, respectively). Although there are more tagged SNPs to consider among models in the 1000-SNP region, the PPs are also more concentrated among these models relative to the large number of SNPs with low evidence of association, meaning fewer models are carried forward at cumulative PP 0.99 for consideration in flashfm.

We provide a wrap-around function that runs single-trait fine-mapping, runs flashfm, constructs SNP groups for both methods, and provides summary results at the SNP and SNP group levels, FLASHFMwithJAM (https://github.com/jennasimit/flashfm/blob/master/R/jamexpanded.corX.R).

We assess robustness of flashfm to misspecified trait correlation by simulating two traits with correlation 0.4 and samples of size 5000, for the original CTLA4 region (1231 SNPs); the traits each have two causal variants, with one shared (Methods). Rankings of the causal variants (using MPP) are compared between the flashfm results using the estimated (non-shifted) trait correlations and those that use trait correlations that are shifted upwards/downwards by 0.1 or 0.2 from the correlation estimate. This region is difficult to fine-map (Methods) and gives a worst case scenario.

There is robustness in the results of flashfm, even when the input trait correlation is shifted upwards/downwards by 0.2. For

**Table 3 Probabilities describing the relationship between flashfm ranks of causal variants when the trait correlation is mis-specified.**

| | Trait 1 (E+G) | | | |
| | Pr(matched ranks) | | Pr(matched or improved ranks) | |
| Trait correlation shift | rs1980422/E | rs3087243/G | rs1980422/E | rs3087243/G |
|---|---|---|---|---|
| −0.2 | 0.870 | 0.923 | 0.960 | 0.950 |
| −0.1 | 0.903 | 0.950 | 0.970 | 0.967 |
| 0.1 | 0.897 | 0.950 | 0.933 | 0.983 |
| 0.2 | 0.793 | 0.900 | 0.857 | 0.947 |
| | Trait 2 (E+H) | | | |
| | Pr(ranks match) | | Pr(matched or improved ranks) | |
| Trait correlation shift | rs1980422/E | rs231775/H | rs1980422/E | rs231775/H |
| −0.2 | 0.850 | 0.913 | 0.940 | 0.977 |
| −0.1 | 0.893 | 0.937 | 0.953 | 0.987 |
| 0.1 | 0.870 | 0.960 | 0.920 | 0.973 |
| 0.2 | 0.760 | 0.897 | 0.837 | 0.927 |

Two traits were simulated to have causal variants E+G and E+H and trait correlation 0.4; sample size is $N = 3000$. Comparisons are made between flashfm results using the estimated trait correlation as input and flashfm results with this trait correlation estimate shifted upwards/downwards by 0.1 or 0.2. The region has 1231 SNPS and was simulated to mimic the LD structure of the CTLA4 region, 2q-204446258-204816382 (GRCh37/hg19). Results are based on 300 replications.

all correlation shifts, the median rankings of causal variants are identical to those based on the correlation estimate - trait 1: 4.5 (E), 1 (G); trait 2: 5.5(E), 2.5 (H). For a more thorough assessment, we examine the probabilities that the rankings match between the shifted and non-shifted input correlation. As rankings that are higher in the shifted analysis are not a negative consequence, we also consider the probability that the shifted analysis ranks are at least as high as those from the original analysis. Our results suggest that flashfm is robust to both positive and negative shifts from the estimated trait correlation (Table 3). The probability that the ranks match between shifted and non-shifted analyses tends to be around 0.90, and ranges from 0.76 to 0.96. Probabilities that the ranking matches or is higher in the shifted analyses over that of the original analysis tend to be around 0.95 and range from 0.837 to 0.987.

**Precision of flashfm is highest among multi-trait methods**. We performed simulations over two regions that mimic the LD structures of IL2RA and CTLA4 and compared the results from flashfm and fastPAINTOR[19]. Both methods use GWAS summary statistics and the SNP correlation matrix to jointly fine-map multiple traits, allowing for multiple causal variants. Flashfm outputs the marginal posterior probabilities (MPP) that a SNP is a causal variant for each trait, whereas fastPAINTOR outputs the MPP that a SNP is causal within the set of traits. Flashfm also outputs the posterior probabilities for multi-SNP models for each trait, whereas fastPAINTOR does not indicate which SNPs are likely joint causal variants, appearing in a model together. For this reason, rather than assessing accuracy in model selection, our comparisons focus on the mean MPPs of the causal variants for each trait and their median rankings, as well as the probability that different proportions of causal variants appear among the top 5 or 10 ranked SNPs for each method. In the IL2RA region we simulated three traits with causal variants A+D, A+C+E, and I and, in the CTLA4 region, two traits with causal variants E+H and E+G (Supplementary Data 1.1, Methods).

In general, flashfm gives higher prioritisation to the causal variants (higher ranking) than fastPAINTOR, and has a higher probability of ranking all causal variants among the top 5 (or 10) SNPs. At shared causal variants, the fastPAINTOR MPPs are similar to those from flashfm in the 3-trait IL2RA simulations, and at some non-shared causal variants flashfm has higher MPPs (Supplementary Data 1.8); e.g. at $N = 5000$, the MPPs for variant I in trait 3 are 0.858 (flashfm) and 0.745 (fastPAINTOR). However, for all causal variants, flashfm tends to have a higher median ranking than fastPAINTOR (Supplementary Data 1.9). For shared variant A, at $N = 5000$, flashfm gives a median ranking of 2 for both traits, while fastPAINTOR gives median ranking 4. As sample size increases, both methods have a similar probability of having at least one causal variant ranked among the top 10 for each trait (Table 4). However, when there are two or three causal variants, the probabilities of ranking all causal variants among the top 10 is noticeably higher for flashfm than fastPAINTOR, even at $N = 5000$ (for all three A +C+E, 0.943 (flashfm) and 0.543 (fastPAINTOR); for both A +D, 0.897 (flashfm) and 0.670 (fastPAINTOR). These probabilities have consistently higher differences for rankings among the top 5, with flashfm consistently higher than fastPAINTOR for two or three causal variants in the top 5 (Supplementary Data 1.10).

Similar patterns are seen in the 2-trait simulations of the CTLA4 region. Flashfm and fastPAINTOR have similar MPPs for causal variants (Supplementary Data 1.11). However, flashfm gives higher prioritisation to all causal variants than fastPAINTOR, even for shared causal variants (Supplementary Data 1.12). For example, at $N = 5000$, flashfm median rankings for shared causal variant E are 4.5 (trait 1) and 5.5 (trait 2), compared to 8.25 (fastPAINTOR). At all sample sizes, the probabilities of at least one causal variant among the top 10, is higher for flashfm than fastPAINTOR, and the flashfm probabilities of both causal variants in the top 10 are twice that of fastPAINTOR (Table 5); at $N = 5000$, the probabilities that both causal variants have rank within the top 10 for trait 1 are 0.867 (flashfm) and 0.457 (fastPAINTOR), and for trait 2 they are 0.783 (flashfm) and 0.353 (fastPAINTOR). Similar patterns are seen for top 5 ranking probabilities, with flashfm also having noticeably higher probabilities of at least 1 causal variant in the top 5, compared to fastPAINTOR (Supplementary Data 1.13).

We also varied trait correlation from 0 to 0.8 for the 3-trait simulations of the IL2RA region at $N = 3000$. Trait correlation does not appear to have a noticeable impact on the MPPs

**Table 4 Comparison of probabilities that causal variants have rank 10 or less, from flashfm and fastPAINTOR, varying sample size.**

| | Trait 1 (A+D) | | | | Trait 3 (I) | |
|---|---|---|---|---|---|---|
| | Pr(1 or more cvs rank <= 10) | | Pr(Both cvs rank <= 10) | | Pr(rank cv <= 10) | |
| N | flashfm | fastPAINTOR | flashfm | fastPAINTOR | flashfm | fastPAINTOR |
| 1000 | 0.787 | 0.393 | 0.130 | 0.057 | 0.943 | 0.557 |
| 2000 | 0.890 | 0.733 | 0.563 | 0.243 | 0.983 | 0.750 |
| 3000 | 0.910 | 0.873 | 0.750 | 0.413 | 0.997 | 0.850 |
| 4000 | 0.937 | 0.917 | 0.837 | 0.560 | 0.997 | 0.903 |
| 5000 | 0.957 | 0.940 | 0.897 | 0.670 | 1 | 0.943 |
| | Trait 2 (A+C+E) | | | | | |
| | Pr(1 or more cvs rank <= 100) | | Pr(2 or more cvs rank <= 10) | | Pr(All 3 cvs rank <= 10) | |
| N | flashfm | fastPAINTOR | flashfm | fastPAINTOR | flashfm | fastPAINTOR |
| 1000 | 0.953 | 0.847 | 0.627 | 0.440 | 0.053 | 0.067 |
| 2000 | 1.000 | 0.963 | 0.910 | 0.747 | 0.497 | 0.253 |
| 3000 | 1.000 | 0.993 | 0.977 | 0.827 | 0.743 | 0.400 |
| 4000 | 1.000 | 0.993 | 0.987 | 0.923 | 0.863 | 0.527 |
| 5000 | 1.000 | 0.997 | 0.993 | 0.930 | 0.943 | 0.543 |

Flashfm tends to have higher probabilities than those from fastPAINTOR, especially for detecting all (multiple) causal variants of a trait. Three traits were simulated to have causal variants A+D, A+C+E, and I and trait correlation is 0.4. Sample size ranges from N = 1000 to 5000. The region has 345 SNPS and was simulated to mimic the LD structure of the IL2RA region, 10p-6030000-6220000 (GRCh37/hg19). Results are based on 300 replications.

**Table 5 Comparison of probabilities that causal variants (cvs) have rank 10 or less, from flashfm and fastPAINTOR, varying sample size.**

| | Trait 1 (E+G) | | | |
|---|---|---|---|---|
| | Pr(1 or more cvs rank <= 10) | | Pr(Both cvs rank <= 10) | |
| N | flashfm | fastPAINTOR | flashfm | fastPAINTOR |
| 1000 | 0.797 | 0.493 | 0.243 | 0.083 |
| 2000 | 0.937 | 0.757 | 0.530 | 0.187 |
| 3000 | 0.987 | 0.823 | 0.677 | 0.303 |
| 4000 | 1.000 | 0.870 | 0.847 | 0.403 |
| 5000 | 0.997 | 0.887 | 0.867 | 0.457 |
| | Trait 2 (E+H) | | | |
| | Pr(1 or more cvs rank <= 10) | | Pr(Both cvs rank <= 10) | |
| N | flashfm | fastPAINTOR | flashfm | fastPAINTOR |
| 1000 | 0.747 | 0.463 | 0.133 | 0.057 |
| 2000 | 0.917 | 0.617 | 0.370 | 0.130 |
| 3000 | 0.960 | 0.737 | 0.600 | 0.237 |
| 4000 | 0.987 | 0.783 | 0.717 | 0.297 |
| 5000 | 0.997 | 0.857 | 0.783 | 0.353 |

For all sample sizes, flashfm consistently has larger probabilities than those from fastPAINTOR. Flashfm has twice the probability of fastPAINTOR for both causal variants to have rank 10 or lower. Two traits were simulated to have causal variants E+G and E+H and trait correlation is 0.4. Sample size ranges from N = 1000 to 5000. The region has 1231 SNPS and was simulated to mimic the LD structure of the CTLA4 region, 2q-204446258-204816382 (GRCh37/hg19). Results are based on 300 replications.

(Supplementary Data 1.14), median rankings (Supplementary Data 1.15), and the probabilities of causal variants in the top 5 (Supplementary Data 1.16) and top 10 (Supplementary Data 1.17).

**Application to cardiometabolic traits in a Ugandan cohort.** Understanding the underlying genetic contributions to cardiometabolic traits is important due to the growing global burden of disability and death attributed to cardiometabolic disorders such as hypertension, coronary artery disease and type 2 diabetes. We considered genetic association signals from 33 cardio-metabolic traits in a Ugandan cohort of 6407 individuals with 45% relatedness of at least second degree[21] (Supplementary Data 1.18 and 1.19,

Supplementary Fig. 4). Based on association signals (p < 1E-6) with at least 2 of the traits, we constructed 56 regions for fine-mapping (Supplementary Data 1.20 and 1.21; Supplementary Fig. 5).

Within the 56 regions the total number of potential causal variants from the top models of FINEMAP was 1147, whereas flashfm reduced this total by 20% to 914 variants (Supplementary Data 2). Among these regions, 52 of them (93%) indicated improved results of flashfm over FINEMAP[5] in terms of either increased PP of the top model, finer resolution of the SNP groups, or both. The median PP increase of flashfm over FINEMAP was 15% (mean 24%) for these regions. The remaining 4/56 regions had concordant SNP groups between

**Table 6 Regions with top models chosen by stepwise (SW), independent fine-mapping and Flashfm where there is a noticeable reduction in SNP group sizes and/or PP of top model.**

| Region | Trait | Stepwise Model | Independent Model (Group Size) | PP | Flashfm Model (Group Size) | PP | Change by Flashfm PP gain | Change by Flashfm Group reduction |
|---|---|---|---|---|---|---|---|---|
| 1:55517883-55674945 (PCSK9,USP24) | LDL | rs11804420/A | A:rs45613943 (4) | 0.5 | A: rs45613943 (3) | 0.63 | 0.13 | A = 25% |
|  | TC | rs11804420/A | A:rs45613943 (4) | 0.6 | A:rs45613943 (3) | 0.76 | 0.12 |  |
| 2:62716187-62887884 (TMEM17) | ALP | rs13403582/B | B:rs7580494 (8) | 0.66 | B:rs6750204 (5) | 0.73 | 0.07 | J = 0 |
|  | PLT | rs765799086/J | J:rs765799086 (1) | 0.46 | B + J:rs765799086 (1) | 0.62 | 0.16 | B = 38% |
| 15:58718136-58742605 (LIPC) | HDL | rs1800588/G | G:rs8033940 (5) | 0.42 | G:rs1800588 (4) | 0.52 | 0.10 | A = 20% |
|  | TG | rs1077835/G | G:rs8033940 (5) | 0.56 | G:rs1800588 (4) | 0.66 | 0.10 |  |
| 16:441156-557188 (LOC100134368, NME4, DECR2, RAB11FIP3) | MCV | rs75167983/L + rs150717215/C + rs147633052/A | L + C + A + B<br>L:rs144739959 (4)<br>C:rs150717215 (2)<br>A:rs147633052 (14)<br>B:rs116567883 (1) | 0.42 | L + C + A + B<br>L:rs75167983 (1)<br>C:rs150717215 (2)<br>A = rs147633052 (14)<br>B = rs116567883 (1) | 0.46 | 0.04 | L = 75%<br>C = 0<br>A = 0<br>B = 0<br>D = 0 |
|  | MCH | rs75167983/L + rs150717215/C + rs147633052/A + rs116567883/B. | L + C + A + B<br>L + C + A + B + D<br>D:rs553374841 (2) | 0.15<br>0.15 | L + C + A + B<br>L + C + A + B + D<br>D = rs553374841 (2) | 0.26<br>0.16 | 0.11<br>0.01 |  |
|  | Bilirubin | L = rs75167983 | L = rs144739959 (4) | 0.45 | L = rs75167983 (1) | 0.63 | 0.18 |  |
| 19:45380937-45441453 (NECTIN2, TOMM40, APOE, APOC1, APOC1P1) | LDL | rs7412/B + rs34215622/V + rs61357706/E + rs429358/L + rs367640607/D2 | E + L + B + V + D2<br>E:rs113152469 (5)<br>L:rs429358 (1)<br>B:rs61679753 (2)<br>V:rs34215622 (1)<br>D2:rs367640607 (2) | 0.41 | E + L + B + V + D2<br>E:rs113152469 (5)<br>L:rs429358 (1)<br>B:rs7412 (1)<br>V:rs34215622 (1)<br>D2:rs367640607 (2) | 0.40 | 0.00 | B = 50%<br>E2 = 25%<br>E = 0<br>L = 0<br>V = 0<br>D2 = 0 |
|  | TC | rs7412/B + rs34215622/V + rs429358/L | E + L + B + V<br>E + L + B + V + D2 | 0.26<br>0.25 | E + L + B + V<br>E + L + B + V + D2 | 0.41<br>0.31 | 0.15<br>0.06 |  |
|  | TG | rs12721054/I + rs5112/X | E + I + Y + E2<br>I = rs12721054 (1)<br>Y = rs7260330 (3)<br>E2 = rs12721051 (4) | 0.42 | E + I + X + E2<br>I = rs12721054 (1)<br>X = rs5112 (1)<br>E2 = rs12721051 (3) | 0.63 | 0.21 |  |
|  | HDL | rs75627662/A | A = rs75627662 (1) | 0.38 | B = rs7412 (1) | 0.42 | 0.03 |  |

Each row summarises results for a single region, defined by chromosome, start and end base-pair position and nearby gene(s). Each cell lists the SNP groups in a model; model A+B indicates all 2-SNP models with one SNP from group A and one SNP from group B. The number of SNPs in each group is given in brackets beside each group in the model. Here we list a representative SNP from each group; rs IDs are from build GRCh37/hg19. The SNPs belonging to each group and their functional annotations are given in Supplementary Data 1.22.

the two methods, but flashfm had a slightly lower PP than FINEMAP (median decrease 0.05%). In 36% of the regions (20/56), flashfm gave a finer resolution than FINEMAP[5], based on the total number of SNPs in the SNP groups of the top model (for each trait) from each method; the median reduction in the total number of SNPs for each trait was 31% with a maximum of 91%. In 88% of the regions (49/56), flashfm had higher confidence than FINEMAP for the top model with a median PP increase of 18% (mean increase 29%); in half of the regions (28/56) flashfm had a noticeable gain with an average PP increase of 52%.

We highlight 5 regions that are improved for both higher confidence in model selection and notable reduction of potential causal variants in the top model of at least one trait (Table 6), and also provide functional annotations for these SNPs, based on HaploReg 4.1[27] and VEP[28] (Supplementary Data 1.22).

Of particular interest is region 19:45380937-45441453 (*PVLR2, TOMM40, APOE, APOC1, APOC1P1*), which shows association to multiple lipid traits: LDL, TC, TG and HDL (Fig. 3). This example highlights the utility of borrowing information between related traits when fine mapping weaker signals. Previous studies have concluded that the LDL association in *APOE* is explained by the missense variants rs7412 and rs429358, which together define *APOE* ε-alleles[29–31]. Both of these SNPs appear in top LDL and TC models by FINEMAP and flashfm. These two methods

identify rs429358 as the only SNP in group L in models for LDL and TC. Flashfm isolates rs7412 as a single SNP in group B, whereas FINEMAP includes an additional SNP in group B, demonstrating that flashfm refines the SNP group of FINEMAP to the correct causal variant.

HDL shows a much weaker association than the other traits (min $p \sim 10^{-9}$), and FINEMAP selects its top SNP rs75627662 as its best candidate. The second strongest SNP in the region for HDL is rs7412 ($r^2$ with rs75627662 is 0.56) and flashfm levers information on LDL and TC to select this SNP as its best candidate for HDL. rs75627662 is an intergenic SNP upstream of *APOE*, while rs7412 encodes a missense variant (Arg158Cys, which is also known as the *APOE* e2 allele). Homozygous carriers for this variant have the condition Type III hyperlipoproteinemia[32] (OMIM 617347), which is characterised by hyperlipidaemia and affected individuals are susceptible to severe coronary artery disease. ApoE is a 34 kDa glycoprotein, initially noted as a component of plasma VLDL and HDL. rs7412 is known to have an effect on the efficacy of Atorvastatin, a statin medication used to prevent cardiovascular disease in those at high risk (ClinVar[33], accession RCV000211178.1); Atorvastatin decreases LDL and triglycerides in the blood and increases HDL. These observations highlight the complex interrelation among LDL, HDL and triglycerides and their regulation and the value of using approaches that capture information from multiple traits.

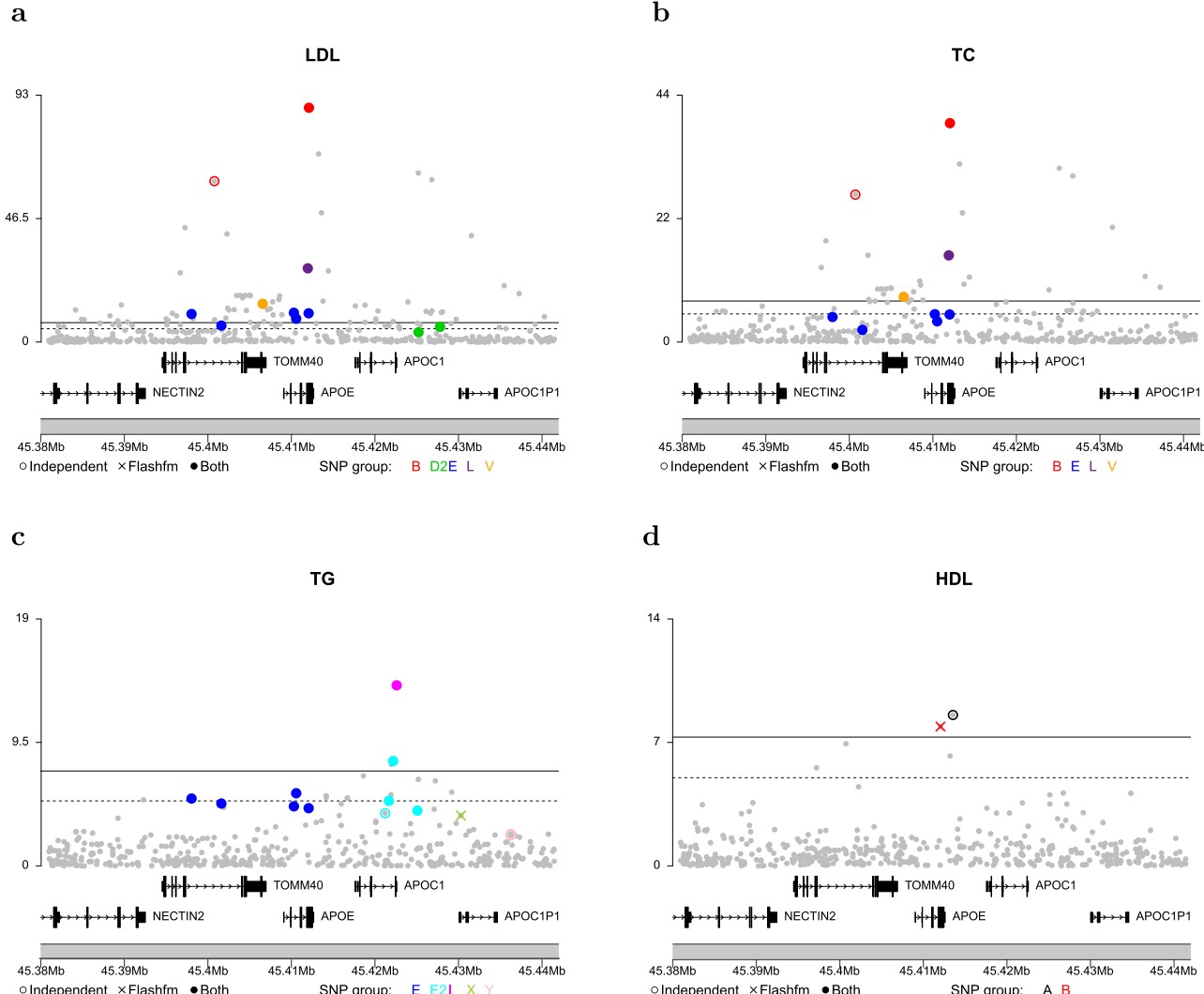

**Fig. 3 Fine-mapping of signals for four lipid traits in region 19:45380937-45441453.** The -log$_{10}$p for SNPs in the top SNP groups for **a** LDL; **b** total cholesterol (TC); **c** triglycerides (TG); **d** HDL are shown for both FINEMAP and flashfm. The two methods agree on a 5-SNP model for LDL (**a**) and a 4-SNP model for TC (**b**). The top model for TG (**c**) has 4 SNPs under both methods but differ in one SNP group; FINEMAP prefers 3-SNP group Y (very near one another so appear as one) and flashfm selected single SNP group X (mean r$^2$ of SNPs in Y with X is 0.315). For HDL (**d**), a different single-SNP model was selected by the two methods; FINEMAP favoured group A, whereas flashfm selected group B. The solid coloured circles show SNPs that belong to the SNP groups constructed by both methods; the empty coloured circles represent SNPs that are only in the FINEMAP SNP group; solid grey circles show all other SNPs in the region. In **c** and **d** an X represents a SNP that appeared in a top model for flashfm and not FINEMAP and empty circles indicate SNPs that appeared in top models for FINEMAP and not flashfm. Position is given according to hg19/build 37. Some of the genes in this region include APOE, APOC1 and TOMM40.

## Discussion

Simultaneous fine-mapping of multiple traits helps in understanding pleiotropic associations by identifying sets of shared potential causal variants that underlie multi-trait associations in the same locus. Jointly fine-mapping traits with flashfm leads to improvements in both accuracy and precision when there are shared causal variants between traits, and leads to similar results to independent fine-mapping when there is no such sharing among traits. Importantly, the flashfm SNP groups are typically a subset of the FINEMAP SNP groups, suggesting that flashfm leverages the information between traits to refine the sets of potential causal variants. This approach only requires GWAS summary statistics, an estimate of the trait covariance matrix, either a genotype reference panel or the covariance matrix and MAFs from a reference panel, and single-trait fine-mapping results (PP for each model). GWAS summary statistics could be

used to approximate the effective sample size for each trait. The input of flashfm makes it readily applicable to meta-analysis results from multiple traits, and the large sample sizes do not increase computational time as only summary-level data are needed.

Flashfm is flexible to accept output from any single-trait fine-mapping approach and, for convenience, we include an expanded version of JAM[4] to obtain single-trait fine-mapping results in the flashfm R package. When using flashfm for the Uganda GWAS of cardiometabolic traits, we used single-trait fine-mapping results from both expanded JAM and FINEMAP[5], which generally resulted in concordant results.

Trait correlation could be influenced by both genetic and environmental correlations, and genetic correlation among traits may be used in place of trait correlation. This is especially recommended when there are GWAS summary statistics from a

cohort that contains related individuals, as the GWAS summary statistics already account for relatedness, provided that a mixed linear model approach had been used. Sodini et al. have shown that, for 17 UK Biobank traits, the genetic correlations calculated from LD score regression[34] are predictive of trait correlation within an independent sample from the same population[35].

In developing flashfm, we use the BIC approximation for BFs[24] to derive an expression for the joint BF, showing that the log(BF) of a joint model for M traits may be expressed as a sum of the marginal log(BF) and a term that depends on the GWAS summary statistics, sample sizes, trait covariance matrix, and LD; GWAS summary statistics are used to approximate the joint SNP effects. As our derivation provides a direct relationship between the joint and marginal BFs and does not disregard trait correlations, LD and joint SNP effects, it should give similar results to multivariate analyses with individual-level genotype data, provided that the same Bayesian framework is used. In its joint fine-mapping, flashfm uses a prior probability that upweights models having a shared causal variant between traits, sharing information between the traits, resulting in improved resolution when traits share causal variants. As flashfm makes use of GWAS summary statistics, it is easily scalable to large biobank-style datasets, whereas individual-level data approaches are not scalable. When there is access to individual-level genotype data, we recommend using this data to calculate the SNP correlation matrix and running a single-trait fine-mapping method, such as JAM or FINE-MAP, as input to flashfm. For smaller sample sizes (e.g., $N = 5000$), users may run the required single-trait fine-mapping with the raw genotype matrix, using the JAMexpanded.multi function of flashfm, followed by flashfm.

Both flashfm and JAM are developed under the assumption of conditional normality and homogeneity of variance. As with any statistical method, departures from these assumptions could potentially produce misleading results. When using summary GWAS data it can be difficult to confirm that modelling assumptions hold, so we urge users to check that these assumptions were tested when the original data were analysed.

Although we find a quick approximation to the joint BF that is based on the marginal BFs, for computational efficiency, we avoid storing these joint BFs in flashfm, and instead use these to approximate the joint PPs (also not stored), which are used to find the trait-adjusted PPs for each trait. Flashfm outputs posterior probabilities for multi-SNP models for each trait, adjusted for information from the other traits, as well as marginal PPs (MPPs) for each SNP under each trait. SNPs with the highest MPPs have more evidence for being causal and the model PPs indicate which combinations of SNPs have evidence of being joint causal variants. We use the MPPs and PPs, together with LD, to construct SNP groups such that SNPs in the same group are in LD and rarely appear in a model together. These groupings allow us to further summarise the SNP-level PP results to SNP group level, which simplifies interpretation and assessing fine-mapping resolution.

As flashfm depends on GWAS summary statistics and model PPs from single-trait fine-mapping, we advise that sample size is assessed in the same way as for a GWAS. When selecting traits to jointly fine-map signals in a region with flashfm, we advise that traits should have a minimum p-value of 1E-6 in the region. Otherwise, fine-mapping will either favour the null model or, if the null model has prior probability of 0, spread the PP over many models. Provided that there is sufficient power to detect a signal in the GWAS and there is a shared causal variant, flashfm will tend to construct SNP groups that are subsets of those from single-trait fine-mapping. In addition, when traits have multiple causal variants, with partial sharing of causal variants, flashfm

shows improvements in prioritising causal variants over another multi-trait fine-mapping method, fastPAINTOR.

As with all existing fine-mapping methods that use summary statistics, inaccurate LD information could reduce the accuracy of the method, either missing causal variants, or inflating evidence for non-causal variants[29]. When a reference panel is the source of the LD matrix for fine-mapping, it must be based on samples of the same ancestry. Benner et al. also show that the size of the reference sample must scale with the GWAS sample size; for a GWAS sample size of 10,000, a reference panel of 1,000 samples is sufficient to estimate LD, whereas a panel of around 10,000 is needed for a GWAS sample size of 50,000.

With the growing availability of biobanks, there are more potential sources for large reference panels and LDstore assists in this as a tool for efficient estimation, storage, and sharing of LD information[29]. LD matrices based on 337,000 British ancestry UK Biobank[36] samples are freely available for download at https://alkesgroup.broadinstitute.org/UKBB_LD[13]; LD matrices for additional ancestry groups (African, Central/South Asian, East Asian, Middle Eastern, Admixed American) within UK Biobank are available for download by the Pan-UKB team at https://pan.ukbb.broadinstitute.org. 2020. Another source for African ancestry LD is the AFR superpopulation of 1000 Genomes[37] that consists of 1,418 samples of African ancestry from both Africa and the United States; data are available for download from http://grch37.ensembl.org/Homo_sapiens/Tools/DataSlicer. As African Americans reflect admixture of people of West and Central-West African descent[38], the AFR superpopulation of 1000 Genomes or the African ancestry cohort from UK Biobank would be an appropriate source of LD for either African or African American samples.

Our simulation studies demonstrated that flashfm tends to give greater improvement in resolution over single-trait fine-mapping when the traits have a common causal variant and a moderate/low correlation between them, though even with highly correlated traits, there is some resolution gain over single-trait fine-mapping. Likewise, the greatest refinement in resolution within the Uganda data tended to be for sets of traits that included at least one trait with a correlation below 0.9 and single-trait fine-mapping top trait models with common SNP groups, which were refined by flashfm. Replication in another data set and downstream functional validation experiments are needed to confirm that the SNPs excluded by flashfm are indeed unlikely to be causal variants. These results promote the joint fine-mapping of traits that are relevant to certain disorders (and have genetic association in the same regions), but not usually considered together; flashfm could not only refine the sets of potential causal variants for the traits, but also reveal new shared causal variants between them.

There are clear advantages to multi-trait fine-mapping with flashfm, and the inclusion of diverse ancestries could lead to further improvements in resolution; where ancestries have common signals, the differences in LD between the ancestries could help pinpoint the causal variant(s)[39]. A multi-ancestry version of flashfm is in progress as a crucial extension to this software, that will allow for multiple reference panels from the different ancestries.

## Methods

**Expanded JAM.** The Joint Analysis of Marginal summary statistics (JAM)[4] facilitates fine-mapping from marginal summary statistics. JAM requires a thinned reference panel such that the genotype matrix does not contain SNPs in high LD and then infers joint LD-adjusted multi-SNP models, highlighting the best SNPs, and combinations of SNPs, via a Bayesian sparse regression framework. Included in the flashfm software, we developed expanded versions of JAM, JAMexpanded.multi (https://github.com/jennasimit/flashfm/blob/master/R/prep.R) and JAMexpandedCor.multi. (https://github.com/jennasimit/flashfm/blob/master/R/jamexpanded.corX.R) that

accounts for all SNPs that were thinned out. Each joint multi-SNP model is expanded by considering all the possible models formed by all the combinations of SNPs in the JAM model, in the same manner as the fine-mapping approach GUESSFM[3] (Supplementary Information 1.3). Then, each of the expanded models is evaluated individually via approximate Bayes' factors. The function JAMexpanded.multi requires a SNP matrix from an in-sample or reference panel and JAMexpandedCor.multi requires a SNP correlation matrix and allele frequencies. FLASHFMwithJAM provides a wrapper that runs JAMexpandedCor.multi, followed by flashfm, as well as construction of SNP groups for both methods and PP summaries by SNP and by SNP group.

**SNP groups**. Rather than reporting results in terms of SNPs, we construct SNP groups using an algorithm based on the group.multi function in GUESSFM (https://github.com/chr1swallace/GUESSFM/blob/master/R/groups.R). SNPs with marginal posterior probability of inclusion > 0.001 were grouped such that SNPs in the same group are in LD - high $r^2$ - and rarely selected together in models (model selection correlation ($r_{model}$) should be negative); both $r_{model}$ and $r^2$ are used so that our SNP grouping is informed by both model posteriors and LD. This algorithm was used in MFM[20] to construct SNP groups based on the single-disease fine-mapping posterior probabilities and these same groups were used to summarise the fine-mapping results from both single-disease and multi-disease (MFM) fine-mapping; using the same groups for both methods allows a convenient comparison of results between the two methods.

As this grouping algorithm makes use of the model posteriors, we extended this grouping algorithm such that the groups are constructed independently for single-trait and multi-trait (flashfm) fine-mapping. That is, SNP groups are constructed for both methods using the same LD information, but groups may differ due to differences in the model PPs between the methods. Since the groups are constructed independently, the new algorithm incorporates a mapping between the group labels of the two sets of groups such that if a flashfm group overlaps a single-trait fine-mapping (STFM) group, the flashfm group label takes the name of the STFM group label. Sometimes a single STFM group overlaps multiple flashfm groups due to the higher precision of flashfm, which could reduce a SFTM group or split it into smaller groups. To account for this, if the STFM group has label A, each flashfm group that intersects A is given label A.1, A.2, etc. to denote that these flashfm groups were from the same larger group in STFM. This algorithm is available as the makeSNPgroups2 function in flashfm (https://github.com/jennasimit/flashfm/blob/master/R/group.multi.R).

**Multiple traits simulations**. Simulations were carried out under a realistic scenario that mimics the MAF and $r^2$ in the IL2RA region (345 SNPs in chromosome 10p-6030000-6220000 (GRCh37/hg19)). This region was selected as it has been previously shown to exhibit a tagging behaviour for causal variants; when there are two causal variants ($A_1$ = rs61839660 and $D_1$ = rs62626317), sometimes a different variant ($B_1$ = rs2104286), that is correlated with both causal variants, is detected as a single causal variant[20]; in this region this tagging behaviour was also observed for two causal variants, $A_1$ = rs61839660 and $C_1$ = rs11594656, jointly tagged by $J_1$ = rs706779. For this region, we generated a population of 100,000 individuals based on the CEU 1000 Genomes Phase 3 data[37] using HapGen2[40]. Details of the LD between $A_1$, $B_1$, $C_1$, $D_1$, and $J_1$ and their MAFs are given in Supplementary Data 1.1. Only variants with MAF > 0.005 were included. An LD plot of this region is given in Supplementary Fig. 1.

For each replication, a random sample of $N$ individuals was selected from the population of 100,000. Causal variants for each trait were selected for a certain disease model, such that trait 1 has two causal variants ($A_1$ and $D_1$) and trait 2 either has two causal variants ($A_1$ and $D_1$); shared $A_1$ with trait 1) or a single distinct causal variant ($C_1$ = rs11594656). Various values for the SNP effects were selected and a multiplicative model was assumed throughout. At each parameter configuration there are 300 replications.

For $M$ traits, the measurement for trait $k$ of individual $j$, $y_{kj}$, is obtained from

$$y_{kj} = \sum_{i=1}^{m_k} \beta_{ik} x_{ij} + \varepsilon_{kj},$$

where $x_{ij}$ is the number of non-reference alleles of variant $i$ for individual $j$ (i.e. genotype score), $\beta_{ik}$ is the effect of causal variant $i$ for trait $k$, $m_k$ is the number of causal variants for trait $k$, and $\varepsilon_{kj}$ is the $k^{th}$ element of the $j^{th}$ multivariate Normal distributed error variable with mean $\mathbf{0}$ and covariance $\Sigma$, which is the covariance matrix of the $M$ traits.

For each replication, Expanded JAM was used for single-trait fine-mapping (SFTM) and used as input into flashfm. We used the makeSNPgroups2 function of flashfm to construct two sets of SNP groups, one set based on SFTM results and one set based on those of flashfm. Results from each method were then summarised based on their coinciding SNP groups, such that a 2-SNP model given by rs61839660 + rs62626317 = $A_1$+$D_1$ is represented by A+D; model A+D consists of all 2-SNP models where one SNP is from group A and the other is from group D. Group labels were assigned according to the variants contained within them, listed as in the beginning of this section for A, B, C, D, and J; if the variant is not captured by a group, then the group is given the label of the variant if it contains a variant that is in LD ($r^2$ > 0.7) with the group variant.

Accuracy of the fine-mapping approaches was evaluated by comparing the mean group model PPs between the two methods. Precision was assessed by comparing the SNP group sizes (number of SNPs in each group) of the groups containing the causal variants for each method; within a replication, for each causal SNP group, the size of the flashfm group was subtracted from the JAM group and the median over the replications was used as a summary. To account for the possibility of a method giving a smaller group size, but not containing the causal variant, we also considered the coverage for each causal variant, defined as the proportion of simulations in which each causal variant is captured by a SNP group. In low power scenarios (e.g. small sample size) the methods may select the correct SNP groups with low PP, and as in a real analysis such SNP groups would not be followed up, we do not consider the SNP group size and coverage for a SNP group if its marginal PP < 0.1 in a replication; all replications are included in the summaries of the model PPs.

**Assessment of multi-trait fine-mapping approaches**. We have also carried out simulations that mimic the MAF and $r^2$ in the CTLA4 region (1231 SNPs in chromosome 2q-204446258-204816382 (GRCh37/hg19)). This region was chosen for its noted difficulty in fine-mapping of multiple autoimmune diseases and we selected causal variants based on previous findings for CTLA4[20]. In particular, causal variants were selected from the SNPs rs1980422, rs231775, and rs3087243, which we denote E, H, and G (Supplementary Data 1.1). An LD plot of this region is given in Supplementary Fig. 3. Simulations in this region and in the IL2RA region were used to evaluate flashfm in comparison to fastPAINTOR[19].

We include traits simulated to have between one and three causal variants and include between two and three traits. In particular, in the IL2RA region we simulate three traits having one, three, and two causal variants, given by rs61839660 + rs62626317 = A+D, rs61839660 + rs11594656 + rs12220852 = A+C+E, and rs706778 = I; these SNPs are plausible, as they have been previously identified for autoimmune diseases[20]. With this setting, we vary sample size from 1000 to 5000, in increments of 1000 and also, at fixed sample size 3000, vary trait correlation from 0 to 0.8, in increments of 0.2. In the CTLA4 region we simulate two traits having two causal variants each: rs1980422+rs3087243 = E+G and rs1980422+rs231775 = E+H with trait correlation 0.4 and sample sizes 1000 to 5000, in increments of 1000.

As fastPAINTOR provides the marginal PP (MPP) of a SNP being a causal variant, and does not output the PPs for joint SNP model configurations, we focus on the MPPs of causal variants and their rankings under each method. For each setting, under flashfm and fastPAINTOR, we compare the following:

1. Mean MPP of SNP being a causal variant, for each causal variant of each trait;
2. Mean and median ranking of each causal variant, for each trait;
3. Probability that at least 1, 2, or 3 (where appropriate) causal variants have rank 5 or less and likewise, 10 or less.

**Evaluation of computation cost and robustness of flashfm**. We have profiled the running time of flashfm, varying the number of traits and the number of SNPs in a locus. This was done in the large CTLA4 region described above, reducing this region to 1000, 500, and 250 SNPs. We simulated 2, 3, and 4 traits within each region. These analyses were run on Intel Skylake 2.6 GHz CPU.

We assess the robustness of flashfm to misspecification of the input trait correlation by simulating two traits in the CTLA4 region, using the same settings as in the multi-trait methods comparison and $N$ = 5000. For 300 replications, we run flashfm using the estimated trait correlation as input, and also with this estimate shifted by −0.2, −0.1, 0.1, and 0.2. For each shift, we calculate the probability that the ranks from the shifted estimate match those of the original analysis, as well as the probability that these ranks from the misspecified correlation match or are higher than the original analysis.

**Fine-mapping of cardiometabolic traits in a Ugandan cohort**. To construct the regions for fine-mapping we used a criterion based on the centimorgan (cM) distance. In particular, for European ancestry populations, fine-mapping regions are often constructed using a boundary of ±0.1 cM around an association signal. When we applied this criterion to the Ugandan cohort, the constructed genome regions were excessively wide. In Park[41] the authors show that a recombination rate of 0.1 corresponds to an LD of 0.4 for European ancestry population, meanwhile for African populations the same level of linkage disequilibrium corresponds to a recombination rate of 0.05. After investigating the dynamics between the recombination rate and the LD, we found that our data required a tighter criterion between 0.03 and 0.05 cM. In order to be conservative and align with the literature, we selected a criterion of ±0.05 cM to define our fine mapping regions (Supplementary Information Section 2). For each region, we also checked a block of 200 SNPs beyond each bound of the region for any SNPs having $r^2$ > 0.4 with the lead SNP in the region. If so, we extend the region to include such SNPs (Supplementary Material Section 2). This procedure resulted in 56 regions with the number of traits by chromosome distribution shown in Supplementary Figs. 5; 21,413,903 SNPs were considered in the analysis, (MAF > 0.005). Most chromosomes had 2 trait-regions which represent 74% of the 56 regions.

Input to flashfm includes single trait fine-mapping results (model posterior probabilities) and single-SNP effect estimates for each trait from the GWAS. We consider two single trait fine-mapping algorithms that use GWAS summary statistics and a reference panel: i) FINEMAP[5] and ii) JAM[4]. FINEMAP is a Bayesian stochastic search algorithm based on summary statistics from GWAS. As there are related samples, the effective sample size, as approximated from the GWAS summary statistics, is used for the number of measurements for each trait; this is easily calculated using the Neff function in flashfm (https://github.com/jennasimit/flashfm/blob/master/R/Neffective.R).

Besides single-trait fine-mapping results, flashfm also needs the single-SNP effect estimates, and a reference (or in-sample) genotype matrix or its covariance matrix. If GWAS summary statistics are available, but not single-trait fine-mapping results, then expanded JAM could be applied to provide input to flashfm. Only SNPs that exist in both the GWAS and the reference panel are carried forward for fine-mapping and this is automated within the flashfm software. Rather than fitting each joint model between traits, we derived an approximation to the joint Bayes' factor (BF) that depends on the marginal BFs, single-SNP effect estimates for each trait, trait covariance matrix, and a reference panel. Based on the single-trait fine-mapping results, flashfm selects the top models from each trait, which are then carried forward for consideration in joint models between the traits. Preferably, models are selected for each trait based on the cumulative posterior probability (cpp; e.g. 0.99 by default), or a maximum number of models may be selected instead.

With exceptions to be clarified on a case-by-case basis, we consider a cpp of 0.99 in our analysis of the Uganda data. For ease of interpretation, as discussed in the SNP Groups section of Methods, main results are presented in terms of SNP groups, rather than individual SNPs; SNPs are grouped such that SNPs within each group are in high LD and rarely appear together in a model. Also, the average LD between groups is lower than 0.6.

For comparison purposes, in addition to results from single-trait fine-mapping from FINEMAP (and JAM in Supplementary Data 1.23) and flashfm (Supplementary Data 3), we also compare our results with the common approach of identifying the best model for the region, stepwise conditional regression[42,43]. In general, FINEMAP and JAM agree on the top model, though there are 6 regions for which JAM could not converge. For this reason, we focus on the more complete FINEMAP results (Supplementary Data 4).

**Reporting summary**. Further information on research design is available in the Nature Research Reporting Summary linked to this article.

## Data availability
Detailed flashfm multi-trait fine-mapping results and FINEMAP single-trait fine-mapping results for the Ugandan cardiometabolic traits are provided in Supplementary Data 3 and 4, respectively; summary fine-mapping results are provided in Supplementary Data 2.pdf. The Uganda GWAS data used in this study are available in the GWAS Catalogue under PubMed ID 31675503 (https://www.ebi.ac.uk/gwas/publications/31675503#study_panel). The Ugandan genotype data are from the European Genome-phenome Archive (EGA) under accession numbers EGAS00001001558 /EGAD00010000965, EGAS00001000545 /EGAD00001001639. The phenotype data used in this study are not under restricted access and requests for access to data may be directed to segun.fatumo@mrcuganda.org. The CEU population 1000Geomes phase 3 haplotype data that were used in our simulations are available from http://grch37.ensembl.org/Homo_sapiens/Tools/DataSlicer.

## Code availability
Our proposed multi-trait fine-mapping method, Flexible and shared information fine-mapping (flashfm), is freely available as an R library at https://jennasimit.github.io/flashfm/ (DOI: 10.5281/zenodo.5522915[44]). Single-trait fine-mapping was performed with FINEMAP 1.4 (http://www.christianbenner.com/), as well as our extended version of JAM (based on JAM from R2BGLiMS; https://github.com/pjnewcombe/R2BGLiMS) that is included in the flashfm package. Custom code for the analysis of the Ugandan data is available at https://github.com/nicolashernandezb/flashfm-analysis. The annotation tools we used are HaploReg v4.1 (https://pubs.broadinstitute.org/mammals/haploreg/haploreg.php) and Ensembl Variant Effect Predictor (VEP) GRCh37 (https://grch37.ensembl.org/info/docs/tools/vep/index.html). We simulated genotype data with hapgen2 (http://mathgen.stats.ox.ac.uk/genetics_software/hapgen/hapgen2.html).

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

## Acknowledgements
J.A. is funded by the MRC (MR/R021368/1) which also supports NH. C.W. is funded by the Wellcome Trust (WT107881) and the MRC (MC_UU_00002/4). P.N. is funded by the MRC (MC UU 00002/9). N.H., P.N., C.W., and J.A. are supported by the NIHR Cambridge BRC (BRC-1215-20014). This work was funded in part by an "Expanding excellence in England" award from Research England to IB. The views expressed are those of the author(s) and not necessarily those of the NHS, the NIHR or the Department of Health and Social Care. This research was funded in whole, or in part, by the Wellcome Trust [WT107881]. For the purpose of Open Access, the author has applied a CC BY public copyright licence to any Author Accepted Manuscript version arising from this submission.

## Author contributions
J.A. initiated and coordinated the project, as well as developed and implemented the flashfm method. C.W. and P.N. contributed to the flashfm method implementation. M.S. provided the Uganda cardiometabolic genetics data. N.H. and J.A. performed statistical analyses. N.H., M.S., I.B., C.W., and J.A. interpreted results. I.B. contributed to expanding the scope of the flashfm method. J.S. cross-validated the flashfm method. J.A., N.H., and C.W. wrote the manuscript with input from all authors.

## Competing interests
The authors declare no competing interests.
