## [Peer Review File · Nature Communications]

The Flashfm Approach for Fine-mapping Multiple Quantitative TraitsREVIEWER COMMENTS

Reviewer #1 (Remarks to the Author):

Hernandez et al., Flashfm: A flexible and shared information fine-mapping approach for multiple quantitative traits.

The manuscript focuses on the problems of 1) multiple traits, 2) related individuals, and 3) missing trait values. These are all important aspects of the overarching problem. Further, the authors are right that many assumptions for existing methods are too restrictive, likely frequently violated (i.e., constrained to the same set of causal variants). The Bayesian approach taken uses GWAS summary statistics and trait covariance matrix to jointly fine map genetic associations for multiple traits. The paper is timely, the theory and derivation is conceptually well done, and the presentation is solid. I do have a few comments that might aid the manuscript.

Introduction:

a) The Introduction is generally well written and covers the necessary information. I would prefer to see more references. For example, the second paragraph of the Introduction is a review of Bayesian methods for fine mapping and would benefit the reader by including more references for the key “best” (by the authors perspective) methods. The three listed are reasonable references but this could be enhanced earlier in the paragraph. If you are hitting a publication limit on references, a few judicious choices might be made.

b) The paragraph on the Uganda samples is overall solid. The choice of both the population studied and the phenotypes are good. I would prefer that the sentence on the phenotypes be reduced. Specifically, I encourage the removal of the parenthetical phrase “which are predicted to become greater in developing countries than that of infectious diseases (e.g., HIV/AIDS)¹².” There is no question that the burden of these diseases are increasing but infectious diseases (e.g., such as Covid-19), including parasites, are a horrible burden in many regions of Africa and should not be minimized. Plenty of public health scholars would argue with you on the respective burden to the population when considering years of life lost and downstream comorbidities. Thus, the choice of phenotypes for an illustrative example is very good, appropriate, but I see no reason for the controversial statement for a good methods paper.

Results

In the beginning of the Results section, you state “multiple quantitative traits that have sample overlap”. Since this is the start of building the analytic framework, please be more specific whether you are proposing a method that requires complete overlap or partial overlap. From the introduction and further reading, I interpret this to mean complete overlap (with potential for missing data which you address elsewhere in the manuscript). Thus, I presume “have complete sample overlap” or “in the same cohort” might orient the reader more quickly to the approach.

Discussion

The discussion is brief but could be more informative.

a) The application of these methods can be for fine mapping applications or could be applied, in windows or other means, genome-wide or across many regions. Your example examined 56 regions, the determination of the number of regions and the total number of SNP can be subjective. The construction of BFs is, by philosophy not a frequentist hypothesis testing perspective, but frankly many people construct BFs and do hypothesis testing like behavior with the posterior odds. Taken together, it might be useful to remind reviewers what are or how best to determine thresholds for making conclusions for the applications you envision being used.

b) The approach is said to show greater power than single trait analyses. It would be useful to guide the reader as per how to estimate that power gain or guidance how to estimate power for grant applications, etc.

c) Can you comment how does the performance of the approach applying summary statistics compares with methods that use the actual genotype data in bivariate analyses? For example, if you have cohorts with genotype data and you are interested in triglycerides and LDL joint modeling. Is this going to be comparably powerful and equally resolving?

d) The methods are developed under the assumption of conditional normality (conditional covariates) and homogeneity of variance. This is valid and fully appropriate. I would have liked some informative comments on what is known about the robustness of the general Bayesian methods, unless you know more specific in your context, for mild to moderate departures from normality (e.g., zero inflation data such as vascular calcium data) and homogeneity of variance assumptions; note, this is not the old ghost criticism about robustness to the prior. I ask because some of the variance component methods are particularly sensitive to heterogeneity of variance in the residuals.

Supplemental material and derivations:

a) I worked through the statistical inference derivations in the Supplemental Section 1, specifically Sections 1 and 2, and I found no typographical “bumps” and the derivation is clear and appears accurate. Some of the algebra is standard likelihood-based inference algebra, but the authors do a nice job of the balance between what is known from an elementary statistical inference class (e.g., Casella and Berger) and their derivatives that are critical to understand in this context. It is well done. I do have a couple of more trivial comments

b) Please write out BIC the first time.

c) I am puzzled why you capitalize most abbreviations (BIC, ABF, BF) but not maximum likelihood estimator (mle). Seems inconsistent, and I have to admit I prefer MLE.

d) In the first sentence you speak of “(transformed so that Normally distributed)”. I presume a more precise statement is that the method assumes the analyst has transformed the phenotypes such that they meet conditional normality and homogeneity assumptions, conditional on covariates. I note this

because so many analysts pick a poor transformation because they transform agnostic to covariates and/or only normality. By doing so they actually reduce power or impede the performance of the methods applied.

e) “Normally”, why capitalize?

f) For those of us with family data, I would value being able to exam using the genetic correlation among phenotypes instead of the global phenotypes. This refinement might have a nice impact on the effectiveness of the method by ignoring the estimated “environmental noise” in the correlations among traits.

As a side comment/opportunity and not a criticism of the current paper, I value MFM methods that allow shared controls. But for many studies, especially very large studies on specialty arrays, there are a mixture of shared controls and study-specific (phenotype-specific) controls and this may be of importance as you don’t want to exclude either subset.

Reviewer #2 (Remarks to the Author):

Herandex et al. have proposed a method, flashfm, to perform fine-mapping of multiple quantitative traits simultaneously. The authors have shown using both simulated and 33 cardiometabolic traits that flashfm outperform single-trait fine-mapping method. Fine-mapping is an important research direction in the field of genetics and human disease. However, I have the following comments.

Major comments:

1. The idea of leveraging multiple traits to improve fine-mapping has been proposed before but the authors fail to compare flashfm with any of the existing methods. The following methods are interesting methods that comparing flashfm can be useful for reader:

a) fastPAINTOR (Kichaev et al. 2017 Bioinformatics)

b) Dap-G (Wen et al. 2016 AJHG)

c) MsCAVIAR (LaPierre et al. 2020 biorxiv)

2. The authors need to compare flashfm results with a couple of fine-mapping methods (e.g., FMF, CAVAIR, CAVIARBF, PolyFUN, DAP-G, SuSiE) after the summary statistics of multiple traits are meta-analyzed using fixed-effect and random effect models.

3. The authors need to show that flashfm is well calibrated in simulated datasets. Reducing the size of fine-mapped variants is a good measure of improvement when methods are well calibrated.

4. The authors need to simulate different genetic architectures to compare flashfm with existing methods. It is important to understand the effect of LD and trait heritability. Fine-mapping in regions with low genetic correlation (LD) is extremely easy while regions with high LD is extremely difficult.

5. The proposed method, flashfm, has a lot of similarity with methods that perform multiple traits colocalization such as HyPrColoc (Foley et al. 2021 Nature Communications) and mcoloc (Giambartolomei et al. 2018 Bioinformatics). The authors need to compare their method and comment on the main distinction with these methods. I agree that these methods are not designed to perform fine-mapping and instead designed for colocalization. However, fine-mapping is an easy outcome of these methods as well.

6. It is important to understand the computation cost of running flashfm. I recommend to profile flashfm running-time while ranging the number of variants (SNPs) in a locus as well as the number of traits used to perform fine-mapping.

7. It is not indicated how the flashfm performs when we have more than two causal variants. I recommend implanting multiple causal variants in the simulated data and compare flashfm with existing methods.

Minor comments:

1. The method section does not explain flashfm model and everything is pushed to Supplementary note. I recommend the authors to move some of the text to the main method section.

2. I recommend the authors to cite the following papers related to fine-mapping: CAVIARBF, PolyFUN, DAP-G, SuSiE, CaVEMaN, KnockoffZoom, and PAINTOR.

RE: **NCOMMS-21-14563A**

We appreciate the constructive comments from the reviewers and feel that our revised version is an improvement over our original manuscript, thanks to their comments. We have given individual responses to each comment below and summarise the significant changes here.

We have extended our simulation study to include comparisons with another multi-trait fine-mapping method, fastPAINTOR - settings include two different regions (LD structures), between one and three causal variants, two and three traits jointly fine-mapped, varying sample size, and varying trait correlation. We have also included a simulation study of 2-4 traits in regions of 250, 500, and 1000 SNPs to profile the running time of flashfm. The software webpage on GitHub (<https://github.com/jennasimit/flashfm>) has also been updated to give more details on how to run flashfm and now includes a new wrapper function that runs single-trait fine-mapping (need GWAS summary statistics and SNP correlation matrix), followed by flashfm, constructs SNP groups for both methods, and summarises the PPs (by SNP and by multi-SNP model) at the SNP-level and SNP group level.

Reviewer #1 (Remarks to the Author):

Hernandez et al., Flashfm: A flexible and shared information fine-mapping approach for multiple quantitative traits.

The manuscript focuses on the problems of 1) multiple traits, 2) related individuals, and 3) missing trait values. These are all important aspects of the overarching problem. Further, the authors are right that many assumptions for existing methods are too restrictive, likely frequently violated (i.e., constrained to the same set of causal variants). The Bayesian approach taken uses GWAS summary statistics and trait covariance matrix to jointly fine map genetic associations for multiple traits. The paper is timely, the theory and derivation is conceptually well done, and the presentation is solid. I do have a few comments that might aid the manuscript.

Introduction:

a) The Introduction is generally well written and covers the necessary information. I would prefer to see more references. For example, the second paragraph of the Introduction is a review of Bayesian methods for fine mapping and would benefit the reader by including more references for the key “best” (by the authors perspective) methods. The three listed are reasonable references but this could be enhanced earlier in the paragraph. If you are hitting a publication limit on references, a few judicious choices might be made.

We have extended our literature review in the Introduction section, in particular, we have added in the references to CAVIARBF and SuSiE:

“Also, fine-mapping methods that make use of GWAS summary statistics (e.g. JAM⁴, FINEMAP⁵, CAVIARBF(Chen et al. 2015), SuSiE(Wang et al. 2020)) assume the specified sample size N relates to independent individuals, whilst the effective sample size after adjustment for relatedness via a linear mixed model, is $<N$.”

And based on the suggested references of the other reviewer, have added the following paragraph:

“Some methods use SNP annotations to improve fine-mapping resolution. A potential caveat of such approaches is that, until the full functional effect of every variant is known, they may bias results towards the biology/function that we already understand. PAINTOR(Kichaev et al. 2014) and DAP-G(Wen et al. 2016) allow for multiple causal variants and integrate either association strength with functional genomic annotation (PAINTOR) or enrichment-based annotations that consider GWAS data from other traits (DAP-G). PolyFUN(Weissbrod et al. 2020) leverages functional annotations to specify prior probabilities for existing fine-mapping methods. The CaVEMaN(Brown et al. 2017) method estimates the probability that the lead SNP for an expression trait is causal for that association, and could assist in SNP prioritisation. KnockoffZoom(Sesia et al. 2020) localizes causal variants at multiple resolutions by testing if a phenotype is independent of all SNPs in a LD block, conditional on the others; it requires individual-level data from unrelated individuals.”

b) The paragraph on the Uganda samples is overall solid. The choice of both the population studied and the phenotypes are good. I would prefer that the sentence on the phenotypes be reduced. Specifically, I encourage the removal of the parenthetical phrase “which are predicted to become greater in developing countries than that of infectious diseases (e.g., HIV/AIDS)¹².” There is no question that the burden of these diseases are increasing but infectious diseases (e.g., such as Covid-19), including parasites, are a horrible burden in many regions of Africa and should not be minimized. Plenty of public health scholars would argue with you on the respective burden to the population when considering years of life lost and downstream comorbidities. Thus, the choice of phenotypes for an illustrative example is very good, appropriate, but I see no reason for the controversial statement for a good methods paper.

Thank you for this comment and we did not intend to minimise the burden of infectious diseases, though understand that it may come across this way. We have removed “which are predicted to become greater in developing countries than that of infectious diseases (e.g., HIV/AIDS)¹².”

Results

In the beginning of the Results section, you state “multiple quantitative traits that have sample overlap”. Since this is the start of building the analytic framework, please be more specific whether you are proposing a method that requires complete overlap or partial overlap. From the introduction and further reading, I interpret this to mean complete overlap (with potential for missing data which you address elsewhere in the manuscript). Thus, I presume “have complete sample overlap” or “in the same cohort” might orient the reader more quickly to the approach.

We thank the reviewer for noticing this area of potential confusion for the reader and we have made this clarification by replacing “have sample overlap” with “have partial sample overlap”.

Discussion

The discussion is brief but could be more informative.

a) The application of these methods can be for fine mapping applications or could be applied, in windows or other means, genome-wide or across many regions. Your example examined 56

regions, the determination of the number of regions and the total number of SNP can be subjective. The construction of BFs is, by philosophy not a frequentist hypothesis testing perspective, but frankly many people construct BFs and do hypothesis testing like behavior with the posterior odds. Taken together, it might be useful to remind reviewers what are or how best to determine thresholds for making conclusions for the applications you envision being used.

We thank the reviewer for this point and have added the following to the Discussion:

“Although we find a quick approximation to the joint BF that is based on the marginal BFs, for computational efficiency, we avoid storing these joint BFs in flashfm, and instead use these to approximate the joint PPs (also not stored), which are used to find the trait-adjusted PPs for each trait. Flashfm outputs posterior probabilities for multi-SNP models for each trait, adjusted for information from the other traits, as well as marginal PPs (MPPs) for each SNP under each trait. SNPs with the highest MPPs have more evidence for being causal and the model PPs indicate which combinations of SNPs have evidence of being joint causal variants. We use the MPPs and PPs, together with LD, to construct SNP groups such that SNPs in the same group are in LD and rarely appear in a model together. These groupings allow us to further summarise the SNP-level PP results to SNP group level, which simplifies interpretation and assessing fine-mapping resolution.”

b) The approach is said to show greater power than single trait analyses. It would be useful to guide the reader as per how to estimate that power gain or guidance how to estimate power for grant applications, etc.

We thank the reviewer for raising this interesting point and we add more guidance on the use of flashfm. Rather than stating an improved power of flashfm over single-trait fine-mapping, we have focused on the gain in accuracy (selecting correct causal variant configuration) and resolution in fine-mapping from flashfm over single-trait fine-mapping, allowing multiple causal variants per trait.

It is difficult to estimate the improvement in accuracy and resolution, but guidance on the use of flashfm is added to the Discussion, as follows:

“As flashfm depends on GWAS summary statistics and model PPs from single-trait fine-mapping, we advise that sample size is assessed in the same way as for a GWAS. When selecting traits to jointly fine-map signals in a region with flashfm, we advise that traits should have a minimum p -value of $1E-6$ in the region. Otherwise, fine-mapping will either favour the null model or, if the null model has prior probability of 0, spread the PP over many models. Provided that there is sufficient power to detect a signal in the GWAS and there is a shared causal variant, flashfm will tend to construct SNP groups that are subsets of those from single-trait fine-mapping. In addition, when traits have multiple causal variants, with partial sharing of causal variants, flashfm shows improvements in prioritising causal variants over another multi-trait fine-mapping method, fastPAINTOR.”

c) Can you comment how does the performance of the approach applying summary statistics compares with methods that use the actual genotype data in bivariate analyses? For example, if

you have cohorts with genotype data and you are interested in triglycerides and LDL joint modeling. Is this going to be comparably powerful and equally resolving?

We thank the reviewer for this comment and have added the following to the Discussion section: “In developing flashfm, we derive an expression for the joint BF, showing that the $\log(\text{BF})$ of a joint model for M traits may be expressed as a sum of the marginal $\log(\text{BF})$ and a term that depends on the GWAS summary statistics, sample sizes, trait covariance matrix, and LD; GWAS summary statistics are used to approximate the joint SNP effects. Considering that flashfm does not ignore trait correlations and also accounts for LD and joint SNP effects, it should give similar results to multivariate analyses with individual-level genotype data, provided that the same Bayesian framework is used. In its joint fine-mapping, flashfm uses a prior probability that upweights models having a shared causal variant between traits, sharing information between the traits, resulting in improved resolution when traits share causal variants. As flashfm makes use of GWAS summary statistics, it is easily scalable to large biobank-style datasets, whereas individual-level data approaches are not scalable. When there is access to individual-level genotype data, we recommend using this data to calculate the SNP correlation matrix and running a single-trait fine-mapping method, such as JAM or FINEMAP, as input to flashfm. For smaller sample sizes (e.g. $N=5000$), users may run the required single-trait fine-mapping with the raw genotype matrix, using the `JAMexpanded.multi` function of flashfm, followed by flashfm.”

d) The methods are developed under the assumption of conditional normality (conditional covariates) and homogeneity of variance. This is valid and fully appropriate. I would have liked some informative comments on what is known about the robustness of the general Bayesian methods, unless you know more specific in your context, for mild to moderate departures from normality (e.g., zero inflation data such as vascular calcium data) and homogeneity of variance assumptions; note, this is not the old ghost criticism about robustness to the prior. I ask because some of the variance component methods are particularly sensitive to heterogeneity of variance in the residuals.

*In general, we do not find Bayesian methods more or less robust to departures from modelling assumptions compared to maximum likelihood methods. The key quantity in our method is the Bayes' factor, which is a ratio of likelihoods under the same distribution but with different parameter values. We cannot quote any formal proof for this, but our experience is that in cases where we have seen substantial errors arise, it has been in the case of data which was highly *unlikely* under either model, ie in the case that the Bayes factor is the ratio of two small quantities (we saw this in a related method due to a mis-coding of the likelihood function, for example). Similarly, we do not expect small non-homogeneity of variance to have a large impact, but again, that is based on our experience using these models rather than any formal proof. Therefore we have added the following comments to the manuscript, drawing attention to this issue, but not making any definitive advice.*

“Both flashfm and JAM are developed under the assumption of conditional normality and homogeneity of variance. As with any statistical method, departures from these assumptions could potentially produce misleading results. When using summary GWAS data it can be difficult to confirm that modelling assumptions hold, so we urge users to check that these

assumptions were tested when the original data were analysed.”

Supplemental material and derivations:

a) I worked through the statistical inference derivations in the Supplemental Section 1, specifically Sections 1 and 2, and I found no typographical “bumps” and the derivation is clear and appears accurate. Some of the algebra is standard likelihood-based inference algebra, but the authors do a nice job of the balance between what is known from an elementary statistical inference class (e.g., Casella and Berger) and their derivatives that are critical to understand in this context. It is well done. I do have a couple of more trivial comments

Thank you for your comment. We are happy to hear that we have reached a good balance for the derivations.

b) Please write out BIC the first time.

Thank you for pointing this out. At the first instance we have added “Bayesian information criterion (BIC)”.

c) I am puzzled why you capitalize most abbreviations (BIC, ABF, BF) but not maximum likelihood estimator (mle). Seems inconsistent, and I have to admit I prefer MLE.

Thank you for pointing this out. All instances of “mle” have been changed to “MLE”.

d) In the first sentence you speak of “(transformed so that Normally distributed)”. I presume a more precise statement is that the method assumes the analyst has transformed the phenotypes such that they meet conditional normality and homogeneity assumptions, conditional on covariates. I note this because so many analysts pick a poor transformation because they transform agnostic to covariates and/or only normality. By doing so they actually reduce power or impede the performance of the methods applied.

Yes, this is correct, and we have made this more precise, changing

“We first suppose that we observe N individuals, each with measurements for M quantitative traits (transformed so that normally distributed); later we relax this so that a subset of individuals may have missing measurements for some of the traits.”

to

“We first suppose that we observe N individuals, each with measurements for M quantitative traits that are transformed to meet conditional normality and homogeneity assumptions, conditional on covariates. Later, we relax this so that a subset of individuals may have missing measurements for some of the traits.”

e) “Normally”, why capitalize?

We have changed “Normally” to “normally”. Out of habit, I usually capitalise distribution names, but realise this may not be standard.

f) For those of us with family data, I would value being able to exam using the genetic correlation among phenotypes instead of the global phenotypes. This refinement might have a nice impact on the effectiveness of the method by ignoring the estimated “environmental noise” in the correlations among traits.

We appreciate this helpful comment, and we thank the reviewer. We agree that when dealing with GWAS summary statistics from family data, the genetic correlation among phenotypes would be a good approximation to the trait correlation because it does not account for any correlation due to environment or other noise. We have added the following to the Discussion: “Trait correlation could be influenced by both genetic and environmental correlations, and genetic correlation among traits may be used in place of trait correlation. This is especially recommended when there are GWAS summary statistics from a cohort that contains related individuals, as the GWAS summary statistics already account for relatedness, provided that a mixed linear model approach had been used. Sodini et al. have shown that, for 17 UK Biobank traits, the genetic correlations calculated from LD score regression(Bulik-Sullivan et al. 2015) are predictive of trait correlation within an independent sample from the same population(Sodini et al.).”

As a side comment/opportunity and not a criticism of the current paper, I value MFM methods that allow shared controls. But for many studies, especially very large studies on specialty arrays, there are a mixture of shared controls and study-specific (phenotype-specific) controls and this may be of importance as you don't want to exclude either subset.

We would like to thank the reviewer for this comment, though we may have misunderstood it. We note that flashfm is intended for joint fine-mapping of signals from quantitative traits measured on the same individuals, allowing for missing measurements. Previously, Multinomial Fine-Mapping (MFM) was developed to deal with joint fine-mapping of signals from diseases, where there are shared controls between the disease studies.

Reviewer #2 (Remarks to the Author):

Herandex et al. have proposed a method, flashfm, to perform fine-mapping of multiple quantitative traits simultaneously. The authors have shown using both simulated and 33 cardiometabolic traits that flashfm outperform single-trait fine-mapping method. Fine-mapping is an important research direction in the field of genetics and human disease. However, I have the following comments.

Major comments:

1. The idea of leveraging multiple traits to improve fine-mapping has been proposed before but the authors fail to compare flashfm with any of the existing methods. The following methods are interesting methods that comparing flashfm can be useful for reader:

- a) fastPAINTOR (Kichaev et al. 2017 Bioinformatics)
- b) Dap-G (Wen et al. 2016 AJHG)
- c) MsCAVIAR (LaPierre et al. 2020 biorxiv)

We agree that a comparison with other multi-trait fine-mapping methods would be useful to the reader and have added in an extensive simulation study that compares flashfm with fastPAINTOR. For a three-trait simulation (traits have one, two, and three causal variants, with one shared between two traits) in one region (IL2RA), we vary sample size from 1000 to 5000

and vary trait correlations from 0 to 0.8. In a different region (CTLA4) we consider sample sizes of 1000 to 5000 for a pair of traits that each have two causal variants, of which one is shared. We have not included Dap-G, as rather than jointly fine-mapping multiple traits, it uses GWAS data from other traits to construct SNP annotations for the fine-mapping of a single trait. As MsCAVIAR is designed for fine-mapping of a single trait from multiple studies, we have not included it in this comparison.

We have added to the Methods section, a section “Assessment of multi-trait fine-mapping approaches” and to the Results section “Precision of flashfm is highest among multi-trait methods”.

2. The authors need to compare flashfm results with a couple of fine-mapping methods (e.g., FMF, CAVAIR, CAVIARBF, PolyFUN, DAP-G, SuSiE) after the summary statistics of multiple traits are meta-analyzed using fixed-effect and random effect models.

Fixed and random effects meta-analysis methods have been historically designed for, and applied to, meta-analyses of effect estimates for the same trait. While random effects meta-analysis does allow for variation in study-specific effect sizes, it still assumes statistical exchangeability across effects; i.e. that they all arise from a common distribution. We would argue that this assumption is inappropriate in the context of effect estimates for different traits, and that neither fixed nor random effects meta-analysis models should be used to pool effect estimates before running single trait fine-mapping methods. Indeed, this was a motivating factor in the development of flashfm, which does not require an assumption of exchangeable effect sizes when modelling shared genetic architecture across traits. We have attempted to better emphasise this point in the paper, by adding the following to the “Flashfm-Conceptual framework” section:

“Flashfm does not require an assumption of exchangeable effect sizes when modelling shared genetic architecture across traits.”

3. The authors need to show that flashfm is well calibrated in simulated datasets. Reducing the size of fine-mapped variants is a good measure of improvement when methods are well calibrated.

We thank the reviewer for this comment that we have addressed by running simulations where one trait has no association signals in the region. The following paragraph has been added to section “Flashfm improves precision over independent fine-mapping”, before describing the results of the simulations of traits that each have at least one causal variant:

“To demonstrate that flashfm is well-calibrated, we considered simulations of two traits such that trait 1 has causal variants A_1 and D_1 and trait 2 has no causal variants in the region. Sample sizes $N=2000$ and 5000 both give nearly identical results between single-trait fine-mapping and flashfm: the PP of the null model for trait 2 when $N=2000$ is 0.960 (single-trait) and 0.957 (flashfm) and for $N=5000$ the null model PPs are 0.966 for both methods. The median difference in SNP group sizes between those constructed from flashfm and those based on independent fine-mapping is zero, and at a given sample size, both methods give the same probabilities that

the SNP groups contain the true causal variants: 0.984 (A; N=2000), 0.991 (D; N=2000), 0.996 (A, D; N=5000).

4. The authors need to simulate different genetic architectures to compare flashfm with existing methods. It is important to understand the effect of LD and trait heritability. Fine-mapping in regions with low genetic correlation (LD) is extremely easy while regions with high LD is extremely difficult.

This is an important point raised, and we have chosen the IL2RA region because we are aware of the difficulty in fine-mapping signals in this region. When simulations are set so that there are two causal variants for a trait, a third distinct SNP that tags both causal variants (moderate LD, $r^2=0.3-0.4$) is sometimes selected instead. We have added Supplementary Figure 1 that illustrates the LD structure in the IL2RA region used for simulations. We have used this region to compare flashfm with fastPAINTOR in simulations of three traits with 1, 2, and 3 causal variants, varying trait correlation and sample size.

We have also added in simulations for the CTLA4 region of chromosome 2, which is a larger region that we found to be difficult to fine-map, based on our previous work. Supplementary Figure 3 illustrates the LD structure. In this new region, we simulated a pair of traits, each with two causal variants, of which one is shared, and used sample sizes from 1000 to 5000. In these simulations, we compare flashfm with fastPAINTOR (new Results section “Precision of flashfm highest among multi-trait methods”)

5. The proposed method, flashfm, has a lot of similarity with methods that perform multiple traits colocalization such as HyPrcoloc (Foley et al. 2021 Nature Communications) and mcoloc (Giambartolomei et al. 2018 Bioinformatics). The authors need to compare their method and comment on the main distinction with these methods. I agree that these methods are not designed to perform fine-mapping and instead designed for colocalization. However, fine-mapping is an easy outcome of these methods as well.

Thank you for this comment and we have added clarification on the difference between flashfm and HyPrcoloc and mcoloc to the Introduction. HyPrcoloc and mcoloc make simplifying assumptions that are not made by flashfm, such as at most one causal variant (both), ignoring trait correlation (both), and traits measured in independent studies (mcoloc). We have changed “When multiple traits have signals in the same region, colocalization is often used to evaluate how likely the traits share a causal variant. In some methods colocalization includes the fine-mapping step of identifying potential shared causal variants, under the simplifying assumption of at most one causal variant for each trait⁶. ”

to

“When multiple traits have signals in the same region, colocalization is often used to evaluate how likely the traits share a causal variant. In some methods colocalization includes the fine-mapping step of identifying potential shared causal variants. HyPrcoloc⁶ and mcoloc make the simplifying assumption of at most one causal variant for each trait. HyPrcoloc ignores trait correlations and is only able to incorporate trait correlations by adjusting the prior configuration probabilities; the authors show that ignoring this adjustment can reduce power to detect a

cluster of colocalised traits. Correlation between traits is not considered by mcoloc, as it requires that all traits are measured in distinct datasets of unrelated individuals.”

6. It is important to understand the computation cost of running flashfm. I recommend to profile flashfm running-time while ranging the number of variants (SNPs) in a locus as well as the number of traits used to perform fine-mapping.

We appreciate this suggestion and agree that it is important to include a profile of the running time. We have added a section “Flashfm is computationally efficient” to the Results section, with the following content:

“We profiled the running time of flashfm, given input from single-trait fine-mapping via expanded JAM that uses the SNP correlation matrix and RAFs (JAMexpandedCor.multi; <https://github.com/jennasimit/flashfm/blob/master/R/jamexpanded.corX.R>), varying the number of SNPs in a region and varying the number of traits in simulated data with 100 replications in each setting. All simulations were done within a region containing CTLA4 (Methods) and we provide the median running times, as well as second and third quartiles (Table 2). For all three region sizes, flashfm tends to run in under one minute when there are two or three traits; at four traits, flashfm tends to run in under 10 minutes.

TABLE 2

Within a given region size, as expected, the time increases with the number of traits. However, there was not an observed increase in time as the region size increases. The region that we continuously reduced initially contained 1231 SNPs and was previously defined for fine-mapping of autoimmune diseases(Asimit et al. 2019): a smaller region for fine-mapping was not selected and then arbitrarily expanded with SNPs that are unlikely to impact the traits (Methods). JAMexpandedCor.multi involves first running JAM single-trait fine-mapping considering multi-SNP models with tag SNPs ($r^2=0.99$), and then expanding these models to include tagged SNPs by interchanging tag SNPs with their tagged SNPs within each model. As the prior probabilities depend on the number of SNPs in a region, the region of 1000 SNPs tended to run faster (4 traits median 168 s) than the smaller regions (4 traits medians 435 s and 583 s for 250 and 500 SNPs, respectively). Although there are more tagged SNPs to consider among models in the 1000-SNP region, the PPs are also more concentrated among these models relative to the large number of SNPs with low evidence of association, meaning fewer models are carried forward at cumulative PP 0.99 for consideration in flashfm.

We provide a wrap-around function that runs single-trait fine-mapping, runs flashfm, constructs SNP groups for both methods, and provides summary results at the SNP and SNP group levels, FLASHFMwithJAM (<https://github.com/jennasimit/flashfm/blob/master/R/jamexpanded.corX.R>).

Number of Traits	250-SNP Region (67 kb)	500-SNP Region (144 kb)	1000-SNP Region (312 kb)

2	2 (1, 7)	5 (2, 15)	5 (1, 16)
3	13 (5, 33)	15 (8, 40)	16 (5, 59)
4	435 (49, 2173)	583 (116, 1790)	168 (32, 740)

Table 2: Median flashfm running time (with second and third quartiles), in seconds.

Flashfm was run using $cpr=0.99$ and single-trait fine-mapping results from JAM, using the extended version (JAMexpandedCor.multi) in the flashfm package. Median time was measured over 100 replications in simulations of 2, 3, and 4 traits having correlation 0.4 and sample size 5000. The regions were subsets of the CTLA4 region 2q-204446258-204816382 (GRCh37/hg19).

And have also provided details in the Methods, under a new section “Evaluation of computation cost of flashfm” with contents:

“We have profiled the running time of flashfm, varying the number of traits and the number of SNPs in a locus. This was done with the large CTLA4 region described above, reducing this region to 1000, 500, and 250 SNPs. We simulated 2,3, and 4 traits within each region. These analyses were run on Intel Skylake 2.6GHz CPU. “

7. It is not indicated how the flashfm performs when we have more than two causal variants. I recommend implanting multiple causal variants in the simulated data and compare flashfm with existing methods.

We thank the reviewer for this helpful comment that improves our presentation of flashfm. We have added in simulations of the IL2RA region that include a trait with 3 causal variants, and have taken this opportunity to also add in a third trait; we simulate traits with causal variants A+D, A+C+E, and I. Both flashfm and fastPAINTOR are applied to these simulations, where both sample size and trait correlation vary. We have added to the Methods section, a new section “Assessment of multi-trait fine-mapping approaches” and a new Results section “Flashfm has highest precision among multi-trait fine-mapping methods”.

Minor comments:

1. The method section does not explain flashfm model and everything is pushed to Supplementary note. I recommend the authors to move some of the text to the main method section.

We thank the reviewer for this suggestion that improves the presentation of flashfm. We have added in a schematic diagram (new Figure 1) and more detail to the “Flashfm – Conceptual framework” section, in particular:

“We find expressions of the $\log(ABF)$ for each of the joint and marginal models by using the approximation based on the Bayesian information criterion (BIC). If the traits are independent,

then the joint ABF of M traits, denoted $ABFM$, is the product of the marginal ABFs. As the traits are correlated the joint ABF is not a simple expression, and we derive the difference, using the log-scale: $D_M = \log(ABF^M) - \sum_{j=1}^M \log(ABF_j)$, which simplifies to a term that depends on GWAS summary statistics, covariance matrix of the traits, and sample sizes; D_M varies for each model configuration. For M traits, $D_M = -N/2(\log|\widehat{C}_M| - \log|C_M|)$, where $|C|$ denotes the determinant of matrix C , C_M is a $M \times M$ matrix with element (i,j) equal to $\text{Cov}(\text{trait } i, \text{trait } j)/\text{Var}(\text{trait } i)$, and \widehat{C}_M is the approximation of C_M . C_M is constant and depends on the trait covariance matrix, whereas \widehat{C}_M is based on the covariance matrix of the residuals specific to each model configuration and is approximated from the GWAS summary statistics, sample sizes, and SNP covariance matrix from a reference panel (Supplementary Information, Section 1.1). This makes the approximation of $ABFM$ computationally feasible: $\log(ABF^M) = \sum_{j=1}^M \log(ABF_j) + D_M$.

2. I recommend the authors to cite the following papers related to fine-mapping: CAVIARBF, PolyFUN, DAP-G, SuSiE, CaVEMaN, KnockoffZoom, and PAINTOR.

We have extended our literature review in the Introduction section, in particular, we have added in the references to CAVIARBF and SuSiE:

“Also, fine-mapping methods that make use of GWAS summary statistics (e.g. JAM⁴, FINEMAP⁵, CAVIARBF⁶ SuSiE⁷) assume the specified sample size N relates to independent individuals, whilst the effective sample size after adjustment for relatedness via a linear mixed model, is $<N$.”

And added the following paragraph with additional references:

“Some methods use SNP annotations to improve fine-mapping resolution. A potential caveat of such approaches is that, until the full functional effect of every variant is known, they may bias results towards the biology/function that we already understand. PAINTOR(Kichaev et al. 2014) and DAP-G(Wen et al. 2016) allow for multiple causal variants and integrate either association strength with functional genomic annotation (PAINTOR) or enrichment-based annotations that consider GWAS data from other traits (DAP-G). PolyFUN(Weissbrod et al. 2020) leverages functional annotations to specify prior probabilities for existing fine-mapping methods. The CaVEMaN(Brown et al. 2017) method estimates the probability that the lead SNP for an expression trait is causal for that association, and could assist in SNP prioritisation. KnockoffZoom(Sesia et al. 2020) localizes causal variants at multiple resolutions by testing if a phenotype is independent of all SNPs in a LD block, conditional on the others; it requires individual-level data from unrelated individuals.”

REVIEWERS' COMMENTS

Reviewer #1 (Remarks to the Author):

The authors have worked to respond to the previous review. I have but a few additional comments.

1. It is stated that the software on GitHub requires summary statistics from GWAS and SNP correlation matrix. If you do not have the SNP correlation matrix directly from the individual genotype data from the GWAS sample and estimate it via other means, how sensitive is the method to misspecification of the SNP correlation matrix? Is it appropriate to use the YRI data to generate the matrix if I am analyzing African Americans? What is the evidence of the robustness of the recommended approach? This should be evaluated.
2. Similarly, if you are using summary statistics, how robust is the method to misspecifying the correlations among the traits?
3. Line 59 and following: The addition of the new paragraph that notes other methods is a positive improvement. However, the statement “a potential caveat of . . . every variant is known” is true for many annotation tools, but other annotations are intrinsic to the DNA itself, such as DNA topology [Neuman et al., *KC. Single-molecule measurements of DNA topology and topoisomerases. J Biol Chem.* 2010 Jun 18;285(25):18967-71; Bettotti et al, *Structure and Properties of DNA Molecules Over The Full Range of Biologically Relevant Supercoiling States. Sci Rep* 8, 6163 (2018); Ainsworth et al., *Intrinsic DNA topology as a prioritization metric in genomic fine-mapping studies. Nucleic Acids Res* 2020].
4. Per individual genotype data versus flashfm summary statistics, you note in your response and in the text that “it should give similar results to multivariate analyses with individual-level genotype data, provided that the same Bayesian framework is used.” However, this is not evaluated, and you do not express experience. Thus, please either do so or make it clear this needs to be evaluated for similar accuracy and precision.

Minor:

Line 535: “Joint” – consistent capitalization for JAM.

Reviewer #2 (Remarks to the Author):

The authors have answered all my concerns.

Reviewer #1 (Remarks to the Author):

The authors have worked to respond to the previous review. I have but a few additional comments. *We thank the reviewer for their comments, which have improved this manuscript.*

1. It is stated that the software on GitHub requires summary statistics from GWAS and SNP correlation matrix. If you do not have the SNP correlation matrix directly from the individual genotype data from the GWAS sample and estimate it via other means, how sensitive is the method to misspecification of the SNP correlation matrix? Is it appropriate to use the YRI data to generate the matrix if I am analyzing African Americans? What is the evidence of the robustness of the recommended approach? This should be evaluated.

Thank you for these helpful comments on the practical aspects of flashfm application. The accuracy of our approach faces the same challenges as single-trait fine-mapping approaches that make use of LD matrices from reference panels and we refer to the extensive studies that have investigated this. We also take this opportunity to refer the reader to publicly available on-line LD resources. To the Discussion section, we have added the following to address these useful questions:

“As with all existing fine-mapping methods that use summary statistics, inaccurate LD information could reduce the accuracy of the method, either missing causal variants, or inflating evidence for non-causal variants. (Benner et al. 2017). When a reference panel is the source of the LD matrix for fine-mapping, it must be based on samples of the same ancestry. Benner et al also show that the size of the reference sample must scale with the GWAS sample size; for a GWAS sample size of 10,000, a reference panel of 1,000 samples is sufficient to estimate LD, whereas a panel of around 10,000 is needed for a GWAS sample size of 50,000.

With the growing availability of biobanks, there are more potential sources for large reference panels and LDstore assists in this as a tool for efficient estimation, storage, and sharing of LD information (Benner et al. 2017). LD matrices based on 337,000 British ancestry UK Biobank(Sudlow et al. 2015) samples are freely available for download at https://alkesgroup.broadinstitute.org/UKBB_LD; LD matrices for additional ancestry groups (African, Central/South Asian, East Asian, Middle Eastern, Admixed American) within UK Biobank are available for download by the Pan-UKB team at <https://pan.ukbb.broadinstitute.org>. 2020. Another source for African ancestry LD is the AFR superpopulation of 1000 Genomes(Consortium and the 1000 Genomes Project) that consists of 1,418 samples of African ancestry from both Africa and the United States; data are available for download from http://grch37.ensembl.org/Homo_sapiens/Tools/DataSlicer. As African Americans reflect admixture of people of West and Central-West African descent (Fatumo et al. 2021), the AFR superpopulation of 1000 Genomes or the African ancestry cohort from UK Biobank would be an appropriate source of LD for either African or African American samples.”

2. Similarly, if you are using summary statistics, how robust is the method to misspecifying the correlations among the traits?

We thank the reviewer for raising this important consideration. To answer this, we ran additional simulations for two traits, where the estimated trait correlation was shifted upwards or downwards by 0.1 or 0.2; this was done over 300 replications. Our result section “Flashfm is computationally efficient” is now called “Flashfm is computationally efficient and robust”, and we add the below there:

“We assess robustness of flashfm to misspecified trait correlation by simulating two traits with correlation 0.4 and samples of size 5000, for the original CTLA4 region (1231 SNPs); the traits each have two causal variants, with one shared (Methods). Rankings of the causal variants (using MPP) are compared between the flashfm results using the estimated (non-shifted) trait correlations and those that use trait correlations that are shifted upwards/downwards by 0.1 or 0.2 from the correlation estimate. This region is difficult to fine-map (Methods) and gives a worst case scenario.

There is robustness in the results of flashfm, even when the input trait correlation is shifted upwards/downwards by 0.2. For all correlation shifts, the median rankings of causal variants are identical to those based on the correlation estimate - trait 1: 4.5 (E), 1 (G); trait 2: 5.5(E), 2.5 (H). For a more thorough assessment, we examine the probabilities that the rankings match between the shifted and non-shifted input correlation. As rankings that are higher in the shifted analysis are not a negative consequence, we also consider the probability that the shifted analysis ranks are at least as high as those from the original analysis. Our results suggest that flashfm is robust to both positive and negative shifts from the estimated trait correlation (Table 3). The probability that the ranks match between shifted and non-shifted analyses tends to be around 0.90, and ranges from 0.76 to 0.96. Probabilities that the ranking matches or is higher in the shifted analyses over that of the original analysis tend to be around 0.95 and range from 0.837 to 0.987.

	Trait 1 (E+G)			
	Pr(matched ranks)		Pr(matched or improved ranks)	
Trait Correlation Shift	rs1980422/E	rs3087243/G	rs1980422/E	rs3087243/G
-0.2	0.870	0.923	0.960	0.950
-0.1	0.903	0.950	0.970	0.967
0.1	0.897	0.950	0.933	0.983
0.2	0.793	0.900	0.857	0.947
	Trait 2 (E+H)			
	Pr(ranks match)		Pr(matched or improved ranks)	
Trait Correlation Shift	rs1980422/E	rs231775/H	rs1980422/E	rs231775/H
-0.2	0.850	0.913	0.940	0.977
-0.1	0.893	0.937	0.953	0.987
0.1	0.870	0.960	0.920	0.973
0.2	0.760	0.897	0.837	0.927

Table 3: Probabilities describing the relationship between flashfm ranks of causal variants when the trait correlation is mis-specified. Two traits were simulated to have causal variants E+G and E+H and trait correlation 0.4; sample size is N=3000. Comparisons are made between flashfm results using the estimated trait correlation as input and flashfm results with this trait correlation estimate shifted upwards/downwards by 0.1 or 0.2. The region has 1231 SNPS and was simulated to mimic the LD structure of the CTLA4 region, chromosome 2q-204446258-204816382 (GRCh37/hg19). Results are based on 300 replications.

We also re-name the Methods section “Evaluation of computation cost of flashfm” to “Evaluation of computation cost and robustness of flashfm” and add the following:

We assess the robustness of flashfm to misspecification of the input trait correlation by simulating two traits in the CTLA4 region, using the same settings as in the multi-trait methods comparison and $N=5000$. For 300 replications, we run flashfm using the estimated trait correlation as input, and also with this estimate shifted by -0.2, -0.1, 0.1, and 0.2. For each shift, we calculate the probability that the ranks from the shifted estimate match those of the original analysis, as well as the probability that these ranks from the misspecified correlation match or are higher than the original analysis.

3. Line 59 and following: The addition of the new paragraph that notes other methods is a positive improvement. However, the statement “a potential caveat of . . . every variant is known” is true for many annotation tools, but other annotations are intrinsic to the DNA itself, such as DNA topology [Neuman et al., KC. Single-molecule measurements of DNA topology and topoisomerases. J Biol Chem. 2010 Jun 18;285(25):18967-71; Bettotti et al, Structure and Properties of DNA Molecules Over The Full Range of Biologically Relevant Supercoiling States. Sci Rep 8, 6163 (2018); Ainsworth et al., Intrinsic DNA topology as a prioritization metric in genomic fine-mapping studies. Nucleic Acids Res 2020].

Thank you for pointing this out and for listing these useful references. We have changed “A potential caveat of such approaches is that, until the full functional effect of every variant is known, they may bias results towards the biology/function that we already understand.”

to

“A potential caveat of such approaches is that they depend on the completeness of the annotation tool. This is not an issue for some annotations that are intrinsic to the DNA itself, such as DNA topology (Neuman 2010) (Bettotti et al. 2018) (Ainsworth et al. 2020). However, most other annotation tools may bias results towards the biology/function that we already understand, until the full functional effect of every variant is known.”

4. Per individual genotype data versus flashfm summary statistics, you note in your response and in the text that “it should give similar results to multivariate analyses with individual-level genotype data, provided that the same Bayesian framework is used.” However, this is not evaluated, and you do not express experience. Thus, please either do so or make it clear this needs to be evaluated for similar accuracy and precision.

We apologise for this lack of clarity and have changed the text to give more evidence for our statement. We would like to emphasise that estimating Bayes’ factors (BF) from GWAS summary data is at the core of all fine-mapping methods, thus a proven strategy for fine-mapping. Within flashfm we re-formulate an existing BF approximation, showing that this approximation can be partitioned into elements that can be estimated in a dataset-specific manner; we do not make any additional approximations.

We have changed

“In developing flashfm, we derive an expression for the joint BF, showing that the $\log(\text{BF})$ of a joint model for M traits may be expressed as a sum of the marginal $\log(\text{BF})$ and a term that depends on the GWAS summary statistics, sample sizes, trait covariance matrix, and LD; GWAS summary statistics are used to approximate the joint SNP effects. Considering that flashfm does not ignore trait correlations and also accounts for LD and joint SNP effects, it should give similar results to multivariate analyses with individual-level genotype data, provided that the same Bayesian framework is used.”

to

“In developing flashfm, we use the BIC approximation for BFs (Wagenmakers 2007) to derive an expression for the joint BF, showing that the $\log(\text{BF})$ of a joint model for M traits may be expressed as a sum of the marginal $\log(\text{BF})$ and a term that depends on the GWAS summary statistics, sample

sizes, trait covariance matrix, and LD; GWAS summary statistics are used to approximate the joint SNP effects. As our derivation provides a direct relationship between the joint and marginal BFs and does not disregard trait correlations, LD and joint SNP effects, it should give similar results to multivariate analyses with individual-level genotype data, provided that the same Bayesian framework is used.”

Minor:

Line 535: “Joint” – consistent capitalization for JAM.

Thank you for catching this and we have corrected the capitalisation for JAM - The Joint Analysis of Marginal summary statistics (JAM)⁴

Reviewer #2 (Remarks to the Author):

The authors have answered all my concerns.

We thank the reviewer for their comments, which have improved this manuscript.